# Leprosy in wild chimpanzees

Kimberley J. Hockings[1,2], Benjamin Mubemba[3,4,24], Charlotte Avanzi[5,6,7,8,24], Kamilla Pleh[3,9,24], Ariane Düx[3,24], Elena Bersacola[1,2,24], Joana Bessa[2,10,24], Marina Ramon[1,24], Sonja Metzger[3,9], Livia V. Patrono[3], Jenny E. Jaffe[3,9], Andrej Benjak[11], Camille Bonneaud[1], Philippe Busso[5], Emmanuel Couacy-Hymann[12], Moussa Gado[13], Sebastien Gagneux[7,8], Roch C. Johnson[14,15], Mamoudou Kodio[16], Joshua Lynton-Jenkins[1], Irina Morozova[17], Kerstin Mätz-Rensing[18], Aissa Regalla[19], Abílio R. Said[19], Verena J. Schuenemann[17], Samba O. Sow[16], John S. Spencer[6], Markus Ulrich[3], Hyacinthe Zoubi[20], Stewart T. Cole[5,21], Roman M. Wittig[9,22], Sebastien Calvignac-Spencer[3] & Fabian H. Leendertz[3,23✉]

Humans are considered as the main host for *Mycobacterium leprae*[1], the aetiological agent of leprosy, but spillover has occurred to other mammals that are now maintenance hosts, such as nine-banded armadillos and red squirrels[2,3]. Although naturally acquired leprosy has also been described in captive nonhuman primates[4–7], the exact origins of infection remain unclear. Here we describe leprosy-like lesions in two wild populations of western chimpanzees (*Pan troglodytes verus*) in Cantanhez National Park, Guinea-Bissau and Taï National Park, Côte d'Ivoire, West Africa. Longitudinal monitoring of both populations revealed the progression of disease symptoms compatible with advanced leprosy. Screening of faecal and necropsy samples confirmed the presence of *M. leprae* as the causative agent at each site and phylogenomic comparisons with other strains from humans and other animals show that the chimpanzee strains belong to different and rare genotypes (4N/O and 2F). These findings suggest that *M. leprae* may be circulating in more wild animals than suspected, either as a result of exposure to humans or other unknown environmental sources.

Leprosy is a neglected tropical disease caused by the bacterial pathogens *M. leprae* and the more recently discovered *Mycobacterium lepromatosis*[8,9]. In humans, the disease presents as a continuum of clinical manifestations with skin and nerve lesions of increasing severity, from the mildest tuberculoid form (or paucibacillary) to the most severe lepromatous type (or multibacillary)[10]. Symptoms develop after a long incubation period ranging from several months to 30 years, averaging 5 years in humans. As a result of sensory loss, leprosy can lead to permanent damage and severe deformity[11]. Although leprosy prevalence has markedly decreased over recent decades, approximately 210,000 new human cases are still reported every year, of which 2.3% are located in West Africa[12]. Transmission is thought to occur primarily between individuals with prolonged and close contact via aerosolized nasal secretions and entry through nasal or respiratory mucosae, but the exact mechanism remains unclear[13,14]. The role of other routes, such as skin-to-skin contact, is unknown.

Leprosy-causing bacteria were once thought to be obligate human pathogens[1]. However, they can circulate in other animal hosts in the wild, such as nine-banded armadillos (*Dasypus novemcinctus*) in the Americas and red squirrels (*Sciurus vulgaris*) in the UK[2,3]. Although initial infection was most probably incidental and of human origin, secondary animal hosts can subsequently represent a source of infection to humans[15–18]. In captivity, nonhuman primates, such as chimpanzees (*Pan troglodytes*)[4], sooty mangabeys (*Cercocebus atys*)[5,6] and cynomolgus macaques (*Macaca fascicularis*)[7], have been known to develop leprosy without any obvious infectious source. However, due to their captive status, it is unclear how they acquired *M. leprae* and whether these species can also contract leprosy in the wild.

Here, we report leprosy infections and their disease course in two wild populations of western chimpanzees (*P. troglodytes verus*) in Cantanhez National Park (CNP), Guinea-Bissau, and in Taï National Park (TNP), Côte d'Ivoire, using a combination of camera trap and veterinary monitoring (Extended Data Fig. 1a and Supplementary Notes 1 and 2). From analyses of faecal samples and postmortem tissues, we identified *M. leprae* as the causative agent of the lesions observed and determined

[1]Centre for Ecology and Conservation, University of Exeter, Penryn, UK. [2]Centre for Research in Anthropology (CRIA – NOVA FCSH), Lisbon, Portugal. [3]Project Group Epidemiology of Highly Pathogenic Microorganisms, Robert Koch Institute, Berlin, Germany. [4]Department of Wildlife Sciences, School of Natural Resources, Copperbelt University, Kitwe, Zambia. [5]Global Health Institute, Ecole Polytechnique Fédérale de Lausanne, Lausanne, Switzerland. [6]Department of Microbiology, Immunology and Pathology, Colorado State University, Fort Collins, CO, USA. [7]Swiss Tropical and Public Health Institute, Basel, Switzerland. [8]University of Basel, Basel, Switzerland. [9]Taï Chimpanzee Project, Centre Suisse de Recherches Scientifiques, Abidjan, Côte d'Ivoire. [10]Department of Zoology, University of Oxford, Oxford, UK. [11]Department for BioMedical Research, University of Bern, Bern, Switzerland. [12]Laboratoire National d'Appui au Développement Agricole/Laboratoire Central de Pathologie Animale, Bingerville, Côte d'Ivoire. [13]Programme National de Lutte Contre la Lèpre, Ministry of Public Health, Niamey, Niger. [14]Centre Interfacultaire de Formation et de Recherche en Environnement pour le Développement Durable, University of Abomey-Calavi, Jericho, Cotonou, Benin. [15]Fondation Raoul Follereau, Paris, France. [16]Centre National d'Appui à la Lutte Contre la Maladie, Bamako, Mali. [17]Institute of Evolutionary Medicine, University of Zurich, Zurich, Switzerland. [18]Pathology Unit, German Primate Center, Leibniz-Institute for Primate Research, Göttingen, Germany. [19]Instituto da Biodiversidade e das Áreas Protegidas, Dr. Alfredo Simão da Silva (IBAP), Bissau, Guinea-Bissau. [20]Programme National d'Elimination de la Lèpre, Dakar, Senegal. [21]Institut Pasteur, Paris, France. [22]Max Planck Institute for Evolutionary Anthropology, Leipzig, Germany. [23]Helmholtz Institute for One Health, Greifswald, Germany. [24]These authors contributed equally: Benjamin Mubemba, Charlotte Avanzi, Kamilla Pleh, Ariane Düx, Elena Bersacola, Joana Bessa and Marina Ramon. ✉e-mail: LeendertzF@rki.de

the phylogenetic placement of the respective strains based on their complete genome sequences.

Chimpanzees at CNP are not habituated to human observers, precluding systematic behavioural observations. Longitudinal studies necessitate the use of camera traps, which we operated between 2015 and 2019. Of 624,194 data files (videos and photographs) obtained across 211 locations at CNP (Extended Data Fig. 1b, Extended Data Table 1 and Supplementary Table 1), 31,044 (5.0%) contained chimpanzees. The number of independent events (images separated by at least 60 min) totalled 4,336, and of these, 241 (5.6%) contained chimpanzees with severe leprosy-like lesions, including four clearly identifiable individuals (two adult females and two adult males) across three communities (Extended Data Fig. 2 and Supplementary Note 2). As with humans, paucibacillary cases in chimpanzees may be present but easily go undetected. Such minor manifestations of leprosy are not reported. All symptomatic chimpanzees showed hair loss and facial skin hypopigmentation, as well as plaques and nodules that covered different areas of their body (limbs, trunk and genitals), facial disfigurement and ulcerated and deformed hands (claw hand) and feet (Fig. 1a–c), consistent with a multibacillary form of the disease. Longitudinal observations showed progression of symptoms across time with certain manifestations similar to those described in humans (such as progressive deformation of the hands) (Extended Data Fig. 2 and Supplementary Videos 1–3). To confirm infection with *M. leprae*, we collected faecal samples and tested them with two nested polymerase chain reaction (PCR) assays targeting the *M. leprae*-specific repetitive element (RLEP) and 18 kDa antigen gene. One out of 208 DNA extracts from CNP was positive in both assays and a second was positive only in the more sensitive RLEP-PCR[19] (Extended Data Table 2, Supplementary Table 2 and Supplementary Note 3). Microsatellite analyses of the two positive samples confirmed that they originated from two distinct female individuals (Supplementary Note 4 and Supplementary Tables 3 and 4). Our results suggest that *M. leprae* is the most likely cause of a leprosy-like syndrome in chimpanzees from CNP.

At TNP, chimpanzees are habituated to the presence of researchers and have been followed daily since 1979. In addition, necropsy samples have been collected from all dead individuals recovered since 2000. In June 2018, researchers first noticed leprosy-like lesions on Woodstock, an adult male chimpanzee from one of the three habituated communities (south) (Extended Data Fig. 1c). The initial small nodules on the ears, lips and under the eye became more prominent and were followed by nodules on the eyebrows, eyelids, nostrils, ears, lips and face. The skin on facial nodules, hands, feet and testicles became hypopigmented and the loss and abnormal growth of nails was observed (Fig 1d–g, Extended Data Fig. 3 and Supplementary Videos 4 and 5). *Mycobacterium leprae* DNA was detected in all samples from June 2018 (Extended Data Table 2, Supplementary Table 2 and Supplementary Note 2). Here, continuous noninvasive detection of *M. leprae* was associated with the onset and evolution of a leprosy-like disease.

Retrospective PCR screening of all chimpanzee spleen samples (*n* = 38 individuals) from the TNP necropsy collection led to the identification of *M. leprae* DNA in two further individuals. An adult female from the same community named Zora, who had been killed by a leopard in 2009, tested positive in both PCR assays. The presence of *M. leprae* DNA was confirmed by PCR in various other organs (Extended Data Table 2). Retrospective analyses of photographs taken in the years before her death showed progressive skin hypopigmentation and nodule development since 2007 (Extended Data Fig. 3). Formalin-fixed skin samples (hands and feet) were prepared for histopathological examination using haematoxylin and eosin as well as Fite-Faraco stains. The skin presented typical signs of lepromatous leprosy characterized by a diffuse cutaneous cell infiltration in the dermis and the subcutis clearly separated from the basal layer of the epidermis (Extended Data Fig. 4a). We detected moderate numbers of acid-fast bacilli (single or in clumps) within histiocytes, indicative of *M. leprae* (Extended Data

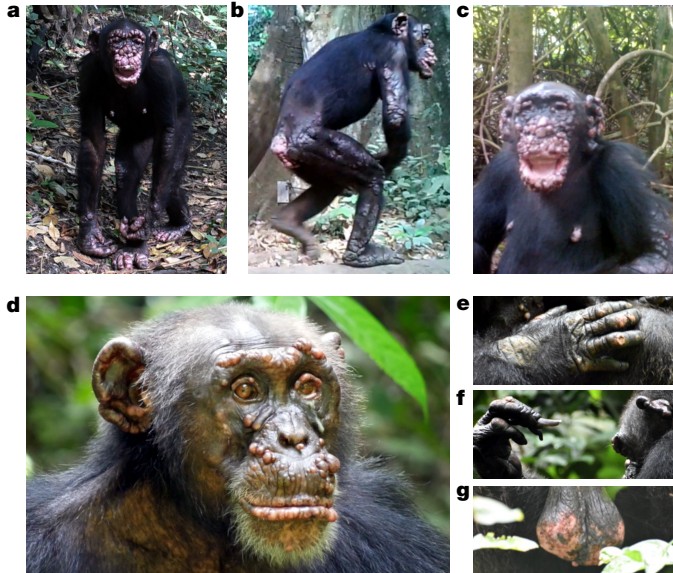

**Fig. 1 | Clinical manifestations of leprosy in three chimpanzees at CNP, Guinea-Bissau and TNP, Côte d'Ivoire. a–c**, Clinical signs of leprosy in two adult female chimpanzees in CNP (images extracted from camera traps). **a**, Rita has large hypopigmented nodules covering the entire body; disfigurement of the face, ears, hands and feet (ulcerated lesions and swelling). **b**, Rita has extensive plaques covering all limbs, with hair loss. **c**, Brinkos has large hypopigmented nodules covering the entire face, with extreme disfigurement of the face and ears, and ulcerated plaques on the arms and the nipples.
**d–g**, Clinical signs of leprosy in an adult male chimpanzee, Woodstock, at TNP.
**d**, Multiple hypopigmented nodules on the ears, brow ridges, eyelid margins, nostrils, lips and the area between the upper lip and the nose.
**e**, Hypopigmentation and swelling of the hands with ulcerations and hair loss on the dorsal side of the joints. **f**, Claw hand with nail loss and abnormal overgrowth of fingernails. **g**, Scrotal reddening and ulceration with fresh blood.

Fig. 4b). As antibodies against the *M. leprae*-specific antigen phenolic glycolipid-I (PGL-I) are a hallmark of *M. leprae* infection in humans[20], we also performed a PGL-I lateral flow rapid test[21] on a blood sample from this individual, which showed strong seropositivity (Extended Data Fig. 4c). Faecal samples collected in the years before Zora's death contained *M. leprae* DNA from 2002 onwards, implying at least 7 years of infection (Extended Data Table 2). In this case, disease manifestations, histopathological findings, serological and molecular data, as well as the overall course of the disease, all unambiguously point towards *M. leprae*-induced leprosy.

To ascertain whether other individuals in the south community of TNP were infected at the time of Zora's death in 2009, cross-sectional screening of contact animals (*n* = 32) was performed by testing all available faecal samples (*n* = 176) collected in 2009 (Supplementary Table 2). Three other chimpanzees were PCR-positive in single samples, including Woodstock. Clinical symptoms of leprosy have not been observed in other individuals, despite daily monitoring of south community members for 20 years and of neighbouring communities for 40 years[22,23]. Considering that, over this period, 467 individuals have been observed, it seems that leprosy is a rare disease with low transmission levels in these chimpanzee communities.

To characterize the *M. leprae* strains causing leprosy in wild chimpanzees and to perform phylogenomic comparisons, we selected DNA extracts that were positive in both the RLEP and the less-sensitive 18-kDa PCR, which indicates relatively high levels of *M. leprae* DNA. For TNP, we selected individuals that were positive in multiple samples. Following targeted enrichment using hybridization capture, samples were subjected to Illumina sequencing. Sufficient *M. leprae* genome coverage

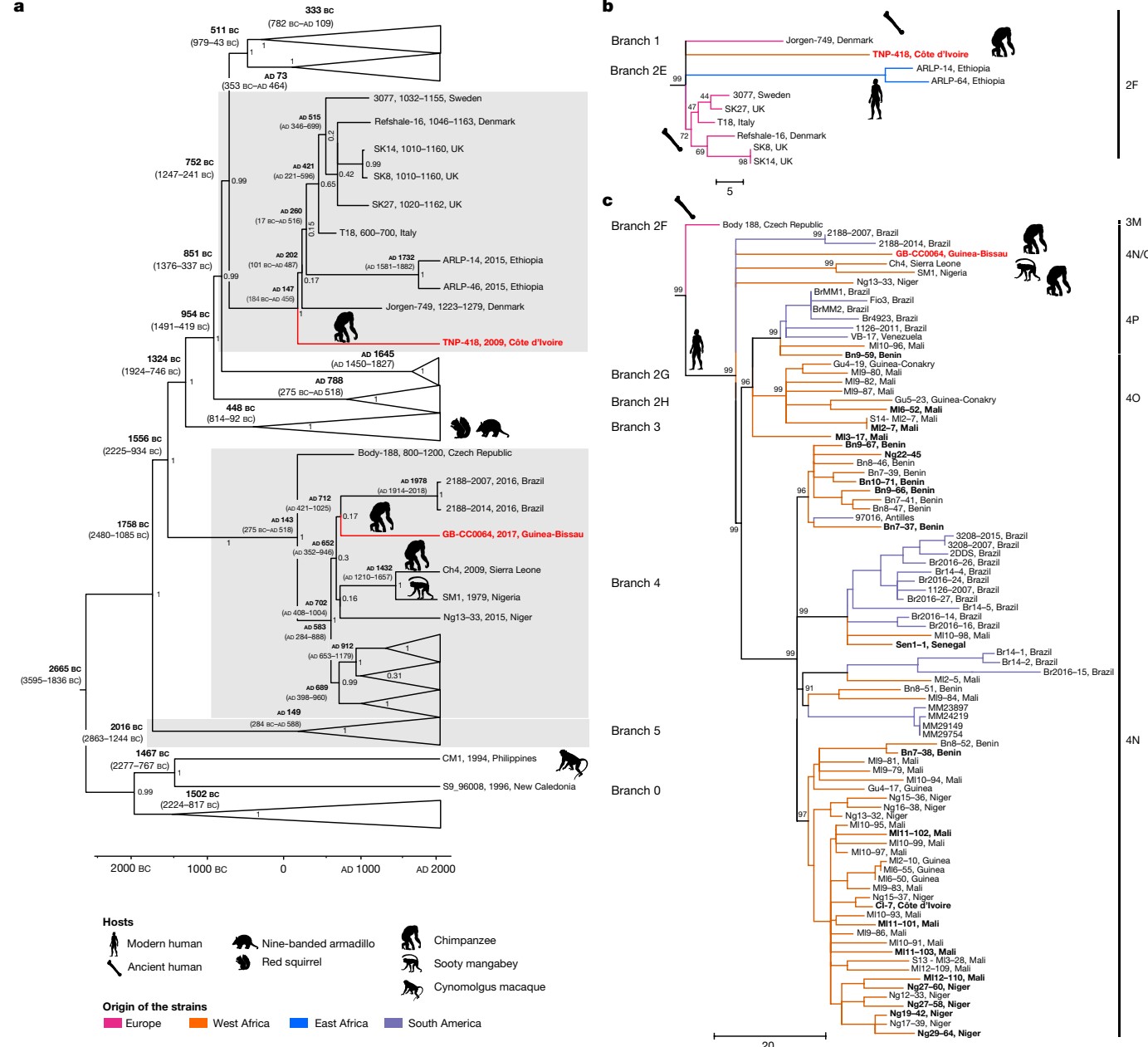

**Fig. 2 | Phylogeny of *M. leprae* strains from human and animal hosts.**
**a**, Bayesian dated phylogenetic tree of 278 *M. leprae* genomes including the two new chimpanzee strains (in bold red). Hypermutated samples with mutations in the *n*th gene were excluded from the analysis. The tree is drawn to scale, with branch lengths representing years of age. Median estimates of node ages are shown in black above branches; 95% HPD intervals are shown in grey. Some *M. leprae* branches are collapsed to increase readability. **b**, Maximum parsimony tree of branch 2F. **c**, Maximum parsimony tree of the branch 4. The tree was initially constructed using 286 genomes (Supplementary Table 6), including 2 new chimpanzee strains (in bold red) and 21 new genomes from West Africa (in bold), 500 bootstrap replicates and *M. lepromatosis* as outgroup. Sites with missing data were partially deleted (80% genome coverage cutoff), resulting in 4,470 variable sites used for the tree calculation. Subtrees corresponding to branches were retrieved in MEGA7[65]. Corresponding genotypes are indicated on the side of each subtree. Samples are binned according to geographical origin as given in the legend. Scale bars (**b**, **c**), number of nucleotide substitutions. Animal silhouettes are available under Public Domain licence at PhyloPic (http://PhyloPic.org/).

was obtained for sample GB-CC064 (Guinea-Bissau) and for Zora (Côte d'Ivoire) with mean depth of 39.3× and 25.8×, respectively (Extended Data Table 2 and Supplementary Table 5). We generated 21 *M. leprae* genomes from human biopsies from five West African countries (Niger, Mali, Benin, Côte d'Ivoire and Senegal) and depth of coverage ranged from 4.7× to 170×. We assembled a dataset that included the genomes generated in this study and all previously available *M. leprae* genomes. Of the total 286 genomes, 64 originated from six West African countries (Extended Data Fig. 5 and Supplementary Note 5).

Bayesian and maximum-parsimony analyses (Extended Data Figs. 6 and 7) place the strain from Guinea-Bissau (GB-CC064) on branch 4, where it clusters outside the standard genotypes 4N, 4O and 4P, but within the so-called 4N/O genotype[24,25] (Fig. 2a, c). This 4N/O genotype is rare and only comprises five *M. leprae* strains; one strain (Ng13-33) from a patient in Niger, two strains (2188-2007 and 2188-2014) obtained from a single patient in Brazil (of 34 strains in Brazil)[26] and two strains from two captive nonhuman primates originating from West Africa (Ch4 and SM1)[25]. The branching order of these five strains and GB-CC064

was unresolved in our analyses, with a basal polytomy suggestive of star-like diversification within this genotype, and within the group comprising all genotype 4 strains (4N/O, 4N, 4P and 4O). Divergence from the most recent common ancestor for this group is estimated to have occurred in the sixth century AD (mean divergence time, 1,437 years ago, 95% highest posterior density (HPD) 1,132–1,736 years ago). The strain that infected Zora in Côte d'Ivoire, designated TNP-418, belongs to branch 2F, within which, the branching order was also mostly unresolved (Fig. 2a, b). The branch is currently composed of human strains from medieval Europe (*n* = 7) and modern Ethiopia (*n* = 2), and this genotype has thus far never been reported to our knowledge in West Africa. Bayesian analysis estimated a divergence time during the second century AD (mean of 1,873 years ago (95% HPD 1,564–2,204 years ago)), similar to previous predictions[27].

Samples from Woodstock did not yield enough Illumina reads to reconstruct full genomes for phylogenomic analysis. However, single-nucleotide polymorphisms (SNPs) recovered from the few available Illumina reads and Sanger sequences derived from PCR products allowed us to assign this second *M. leprae* strain from Côte d'Ivoire to the same genotype as TNP-418 (Supplementary Note 5). Overall, phylogenomic analyses show that *M. leprae* strains in chimpanzee populations at CNP and TNP are not closely related.

The finding of *M. leprae*-induced leprosy in wild chimpanzee populations raises the question of the origin(s) of these infections. *Mycobacterium leprae* is considered a human-adapted pathogen and previous cases of leprosy affecting wildlife were compatible with anthroponosis. Therefore, the prime hypothesis would be human-to-chimpanzee transmission. Potential routes of transmission include direct (such as skin-to-skin) contact and inhalation of respiratory droplets and/or fomites, with the assumption that, in all cases, prolonged and/or repeated exposure is required for transmission[11]. Chimpanzees at CNP are not habituated to humans and are not approached at distances that would allow for transmission via respiratory droplets. Although these chimpanzees inhabit an agroforest landscape and share access to natural and cultivated resources with humans[28], present-day human–chimpanzee direct contact is uncommon. The exact nature of historic human–chimpanzee interactions at CNP remains, however, unknown. For example, robust data on whether chimpanzees were kept as 'pets' or were hunted for meat are lacking. Long-term human–chimpanzee coexistence in this shared landscape makes humans the most probable source of chimpanzee infection. However, multiple individuals from several chimpanzee communities across CNP show symptomatic leprosy demonstrating that *M. leprae* is now probably transmitted between individuals within this population.

At TNP, the south chimpanzee community is distant from human settlements and agriculture. Human-to-animal transmission of pathogens has been shown at TNP[29,30] but involved respiratory pathogens (pneumoviruses and human coronavirus OC43) that transmit easily and do not require prolonged exposure. In addition, *M. leprae* is thought to be transmitted from symptomatic humans[31] and no cases of leprosy have been reported among researchers or local research assistants. Although a human source is impossible to rule out, low human contact coupled with the rarity of the *M. leprae* genotype detected in TNP chimpanzees among human populations in West Africa suggests that recent human-to-chimpanzee transmission is unlikely. This is supported by the absence of drug-resistant mutations (Supplementary Note 6). The relatively old age of the lineage leading to the chimpanzee strain at TNP nevertheless raises the possibility of an ancient human-to-chimpanzee transmission. However, the human population density 1,500–2,000 years ago was probably even lower than it is currently, making this unlikely. If such an ancient transmission had occurred and the bacterium had persisted for a long time in chimpanzees, it should have spread more broadly as observed in *M. leprae*-infected squirrels and armadillos[3,16,17]. Therefore, an ancient human-to-chimpanzee transmission is not the most plausible mechanism to explain the presence of *M. leprae* in chimpanzees at TNP.

These findings may be better explained by the presence of a nonhuman leprosy reservoir. As chimpanzees hunt frequently, transmission may originate from their mammalian prey[32]. Nonhuman primates are the most hunted prey at TNP[33] and are hunted at CNP (Supplementary Note 3). Chimpanzees also consume other mammalian prey such as ungulates. Notably, this scenario assumes that the animal host range of *M. leprae* is even broader than is currently known. Perhaps more intriguingly, an environmental source may be at the origin of chimpanzee infections. Other mycobacteria can survive in water, including *M. ulcerans* and other non-tuberculous mycobacteria[34,35], and molecular investigations have reported that *M. leprae* can survive in soil[36]. Experimental data also show that *M. leprae* multiplies in amoebae[37], arthropods[38] and ticks[39], which could contribute to the persistence of the bacteria in the environment. Testing these hypotheses will require thorough investigation of the distribution of *M. leprae* in wildlife and the environment and so shed light on the overall transmission pathways of the pathogen.

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

# Methods

## Study sites

Observational study and sample collections were performed at CNP in southern Guinea-Bissau and TNP in western Côte d'Ivoire (Extended Data Fig. 1a). CNP (1,067 km²) comprises the Cubucaré peninsula in the sector of Bedanda, with the northeast of the park bordering the Republic of Guinea. The landscape at CNP consists of a mosaic of mainly mangroves, subhumid forest patches, savannah grassland and woodland, remnant forest strips dominated by palm groves as well as agriculture[40]. There are approximately 200 villages and settlements within the borders of the park, with an estimated human population of 24,000 individuals who comprise several ethnic groups[41]. Chimpanzees are not hunted for consumption within CNP due to local cultural beliefs and taboos[42] but are sometimes killed in retaliation for foraging on crops[43,44]. There is a minimum of 12 chimpanzee communities at CNP[41], all unhabituated to researchers, with approximately 35–60 individuals per community[45,46]. Numerous other wildlife taxa inhabit CNP, including six other nonhuman primate species[41,47].

The TNP (5,082 km²) consists of an evergreen lowland rainforest and is the largest remaining primary forest fragment in West Africa. It is home to a wide range of mammals that include 11 different nonhuman primate species[48,49]. There are no settlements or agricultural areas inside the National Park. As of March 2021, the three habituated communities, north, south and east, comprised 22, 37 and 32 individuals, respectively, although community sizes have varied over time. Systematic health monitoring of these communities has been ongoing since 2000[23].

## Longitudinal observations and health monitoring

At CNP, camera traps (Bushnell Trophy Cam models 119774, 119877 and 119875) were deployed at 211 locations, including across different habitat types (forest, mangrove-forest edge and orchards) within the home range of 8 of the 12 putative chimpanzee communities (Supplementary Table 1). Camera traps were set up over six data collection periods from 2015 to 2019 (Extended Data Table 1). Targeted camera traps were deployed to record and monitor chimpanzee behaviour and disease occurrence. To maximize the chances of recording specific behaviours and to identify leprosy-like symptoms in individuals, targeted camera traps were set up in locations that chimpanzees were known to use most often, sometimes in clusters, precluding uniform survey designs. Targeted camera traps were set up in video mode and were active 24 h per day. When triggered, targeted cameras recorded 10 to 60 s of video with a minimum interval of 0.6 s or 2 s, depending on the camera trap model. Furthermore, systematically placed camera traps were used to obtain measures of wildlife occurrence and habitat use across the heterogeneous landscape[41]. Systematic camera traps were deployed across central CNP, at a minimum distance of 1 km between sampling points, as well as within the home range of one chimpanzee community (Caiquene-Cadique) and were spaced at least 500 m from one another. The camera traps pointed towards animal paths (often chimpanzee paths), small human paths also used by wildlife and other areas presenting signs of animal activity. Systematic camera traps were set up to record three consecutive photographs when triggered. The GPS coordinates, habitat type, date, time and site description were recorded when setting up individual camera traps (targeted and systematic). Opportunistic observations of chimpanzees at CNP were made in 2013, during which chimpanzees were photographed and/or filmed using digital cameras.

Chimpanzees at TNP are fully habituated to human observers and all individuals in the habituated communities are individually identified. Behavioural and health monitoring of chimpanzees at TNP involves daily observation of habituated individuals by an interdisciplinary team comprising primatologists and veterinarians; investigations of wildlife mortality causes through necropsies on all animal carcasses found in the research area; and the collection of noninvasive samples such as faecal samples, laboratory investigations and the communication of the results to the park management for corrective and preventive measures[22]. Abnormalities in behaviour or clinical signs of disease are immediately reported and followed by detailed observation by the on-site veterinarian. To reduce the risk of transmission of human diseases to the chimpanzees, stringent hygiene measures have been put in place, including an initial 5-day quarantine for observers, keeping a distance of at least 7 m and obligatory wearing of masks, with only healthy observers allowed to work in the forest[50,51].

## Faecal and necropsy sample collection

At CNP, chimpanzee faecal samples were collected between July 2017 and December 2018. The date and putative chimpanzee community were recorded for each faecal sample. As defecation was rarely observed and to prevent the collection of redundant samples from the same individual, we avoided multiple samples found under the same chimpanzee nest and paid special attention if multiple samples were found in proximity on trails[45,52,53]. All samples were collected with the aid of a wooden spatula and stored at ambient temperature in 15-ml tubes containing NAP buffer[54]. All samples were sent to the Robert Koch Institute for laboratory analysis. Even though chimpanzee faeces are easily distinguishable from those of other species and were found in areas where chimpanzees had recently been present with associated signs such as feeding remains or knuckle prints, we genetically confirmed the presence of chimpanzee DNA in faecal samples that tested positive in either of the *M. leprae* PCRs or the mammal PCR for diet analysis (Supplementary Note 3).

At TNP, the long-term health monitoring programme includes continuous collection of faecal and urine samples from known adult chimpanzees. Faeces are collected right after defecation, transferred to 2-ml cryotubes with the aid of a plastic spatula and frozen in liquid nitrogen the same day. A full necropsy is systematically performed on chimpanzees found dead by the on-site veterinarian. Necropsies follow a standardized biosafety protocol due to the occurrence of anthrax, Ebola and monkeypox in the area. This includes the use of full personal protective equipment and rigorous disinfection measures. Tissue samples of several internal organs are taken if the state of carcass decomposition allows. After collection, all samples are first stored in liquid nitrogen and subsequently shipped on dry ice to the Robert Koch Institute for analyses.

## DNA extraction from faeces and necropsy samples

DNA extractions were performed at the Robert Koch Institute in a laboratory that has never been used for molecular *M. leprae* investigations. DNA was extracted from faecal and necropsy samples using the GeneMATRIX stool DNA purification kit (EURx) and the DNeasy Blood and Tissue kit (QIAGEN), respectively, following the manufacturers' instructions. Extracted DNA was then quantified using the Qubit dsDNA HS Assay kit (Thermo Fisher Scientific) and subsequently stored at −20 °C until further use.

## Genetic identification of samples from infected chimpanzees at CNP

To determine whether faecal samples positive for *M. leprae* belonged to one or two individuals at CNP, we amplified chimpanzee DNA at 11 microsatellite loci and one sexing marker[55]. Owing to the small quantity of starting DNA, not all loci were amplified and in some cases the amplification quality was low, affecting our ability to confidently interpret allele peak profiles (for example, sample GB-CC064 failed to amplify for 5 out of the 11 loci) (Supplementary Note 4).

## Molecular screening of *M. leprae* in faecal and necropsy samples

*Mycobacterium leprae* DNA was searched for using two nested PCR systems targeting the distinct but conserved repetitive element RLEP

and the 18-kDa antigen gene as previously described (Extended Data Table 3). As 37 copies of RLEP are present in the *M. leprae* genome, this assay is considered to be more sensitive than 18 kDa, for which there is only a single copy. To prevent contamination at the laboratory at the Robert Koch Institute and to enable us to identify whether it occurs, we followed these procedures: (1) separate rooms were used for preparation of PCR master mixes and the addition of DNA in the primary PCR; (2) the addition of the primary PCR product in the nested PCR in another separate room; and (3) dUTPs were used for all PCRs instead of dNTPs. For both assays, primary PCRs were performed in 20-μl reactions: up to 200 ng of DNA was amplified using 1.25 U of high-fidelity Platinum Taq polymerase (Thermo Fisher Scientific), 10× PCR buffer, 200 μM dUTPs, 4 mM MgCl$_2$ and 200 nM of both forward and reverse primers. The thermal cycling conditions for the primary and nested PCRs were as follows: denaturation at 95 °C for 3 min, followed by 50 cycles of 95 °C for 30 s, 55 °C (18 kDa primers) or 58 °C (RLEP primers) for 30 s, and 72 °C for 1 min as well as an elongation step at 72 °C for 10 min. For nested PCRs, 2 μl of a 1:20 dilution of the primary PCR product was used as a template. Molecular-grade water was used as a template-free control. PCR products were visualized on a 1.5% agarose gel stained with GelRed (Biotium). Bands of the expected size were purified using the Purelink Gel extraction kit (Thermo Fisher Scientific). Both RLEP and 18-kDa nested PCR products are too short for direct Sanger sequencing. Therefore, fusion primers (primary PCR primers coupled with M13F and M13R primers) (Extended Data Table 3) were used for further amplification of the cleaned PCR products, applying the same conditions as in the primary PCR, but running only for 25 cycles. The resulting extended PCR products were then enzymatically cleaned using the ExoSAP-IT PCR Product Cleanup assay (Thermo Fisher Scientific) and Sanger sequenced using M13 primers. Resulting sequences were compared to publicly available nucleotide sequences using the Basic Local Alignment Search Tool (BLAST)[56].

## Histopathology

To further confirm the infection, skin samples were sent to the German Primate Center in Göttingen, Germany for histopathological analyses. Samples were immersion-fixed in 10% neutral-buffered formalin, embedded in paraffin and stained with standard haematoxylin and eosin using the Varistain Gemini staining automat (Thermo Fisher Scientific). Samples were also stained with Fite-Faraco stain for the identification of acid-fast bacilli.

## Serology

A whole-blood sample from Zora collected during the necropsy in 2009 was tested for the presence of the *M. leprae*-specific anti-PGL-I antibodies using a chromatographic immunoassay developed for use with human blood following the instructions provided by the test manufacturers with a 1:10 diluted whole-blood sample. This rapid lateral flow test was produced by R. Cho using the synthetic ND-O-BSA antigen with financial support of the NIH/NIAID Leprosy Research Materials contract AI-55262 at Colorado State University. Test results were interpreted at 5 and 10 min. Human serum from a patient with multibacillary leprosy donated by J. S. Spencer, Colorado State University, was used as a positive control. Whole blood collected during the necropsy of a chimpanzee (Olivia) at TNP who died of acute respiratory disease in 2009 was used as a negative control.

## Library preparation, genome-wide capture and high-throughput sequencing for nonhuman primate samples

Selected *M. leprae*-positive faecal and necropsy samples (Supplementary Table 2) were converted into dual-indexed libraries using the NEBNext Ultra II DNA Library Prep kit (New England Biolabs)[57,58]. To reconstruct whole genomes, libraries were target-enriched for *M. leprae* DNA using in-solution hybridization capture with 80-nt RNA baits designed to cover the whole *M. leprae* genome (twofold tiling;

design can be shared upon request to the corresponding author) and following the myBaits protocol as previously described[25]. Around 1.5 μg of each DNA library was captured in single or pooled reactions. Two rounds of 24-h hybridization capture were performed followed by a post-amplification step for each using the KAPA HiFi HotStart Library Amplification kit with 12 to 16 cycles to generate around 200 ng of enriched library per sample. Finally, enriched libraries were purified using the silica-based MinElute reaction cleanup kit (QIAGEN) followed by quantification with the KAPA library quantification kit (Roche). Libraries were then normalized and pooled across sequencing lanes on an Illumina NextSeq 500 for sequencing with a mid-output kit v.2 for 300 cycles (Illumina).

## Sample collection, DNA extraction, library preparation, genome-wide capture and high-throughput sequencing of human specimens

Samples (skin biopsies or DNA extracts) from patients with leprosy from five West African countries who had a positive bacillary index (Niger (*n* = 5), Mali (*n* = 8), Benin (*n* = 6), Côte d'Ivoire (*n* = 1) and Senegal (*n* = 1)) were obtained from the respective National Leprosy Control Programmes in the framework of the leprosy drug-resistance surveillance programmes or from previous investigation[59].

DNA was extracted from skin biopsies using the total DNA extraction method as described previously[60]. DNA was quantified with a Qubit fluorometer using the Qubit dsDNA BR Assay kit (Thermo Fisher Scientific) before library preparation. DNA libraries were prepared using the KAPA Hyper Prep kit (Roche) as per the manufacturer's recommendation using KAPA Dual-Indexed Adapter (Roche) followed by in-solution capture enrichment with 80-nt RNA baits with 2× tiling density for 48 h at 65 °C as described previously[60]. Post-capture amplification was performed with seven cycles. Enriched libraries were purified using a 1× ratio of KAPA Pure beads (Roche) followed by quantification with the KAPA library quantification kit (Roche) and quality control of the fragment with the Agilent 2200 TapeStation (Agilent Technologies). Libraries were then normalized and pooled across sequencing lanes on an Illumina NextSeq 500 for sequencing with a high output kit v.2 for 75 cycles (Illumina).

## Genomic data analysis

Raw reads were processed as described elsewhere[24]. Putative unique variants of GB-CC064 and TNP-418 strains were manually checked and visualized using the Integrative Genomics Viewer[61].

## Genome-wide comparison and phylogenetic tree

SNPs of the two newly sequenced genomes from chimpanzees were compared to the 263 publicly available *M. leprae* genomes[25,60,62–64] (Supplementary Table 6) and 21 new genomes from West African countries (Supplementary Note 5). Phylogenetic analyses were performed using a concatenated SNP alignment (Supplementary Table 7). Maximum-parsimony trees were constructed in MEGA7[65] with the 286 genomes available (Supplementary Table 5) using 500 bootstrap replicates and *M. lepromatosis*[66] as outgroup. Sites with missing data were partially deleted (80% genome coverage cutoff), resulting in 4,470 variable sites used for the tree calculation.

## Dating analysis

Dating analyses were performed using BEAST2 (v.2.5.2)[67] as described previously[24] with 278 genomes and an increased chain length from 50 to 100 million. In brief, concatenated SNPs for each sample were used for tip dating analysis (Supplementary Table 7). Hypermutated strains and highly mutated genes associated with drug resistance (in yellow, Supplementary Table 7) were omitted[24,60], manual curation of the maximum parsimony and BEAST input file was conducted at the positions described in Supplementary Table 9 for GB-CC064 and TNP-418. Sites with missing data as well as constant sites were included in the

analysis, as previously described[24]. Only unambiguous constant sites (loci where the reference base was called in all samples) were included.

## PCR genotyping of insufficiently covered *M. leprae* genomes from positive chimpanzees

The genome coverage for the strain infecting Woodstock was low. To be able to determine the genotype, we identified specific variants from the genome-wide comparison of TNP-418 (the strain infecting Zora, an individual from the same social group) with other strains from branch 2F (Supplementary Table 9). Variants were manually checked and visualized in the partially covered genome from the strain infecting Woodstock using IGV software (Supplementary Table 10). Two variants not covered by high-throughput sequencing data were also selected for specific PCR screening. Primers were designed using the Primer3 web tool (http://bioinfo.ut.ee/primer3-0.4.0/) based on Mycobrowser sequences[68] and are described in Extended Data Table 3. All PCR conditions were the same as in the *M. leprae* screening PCRs except for the primer sets and associated annealing temperatures.

## Ethical oversight

For chimpanzees, all data were collected in accordance with Best Practice Disease and Monitoring Guidelines developed by the Section on Great Apes, IUCN SSC Primate Specialist Group (IUCN SSC PSG SGA). The collection of samples was noninvasive. All proposed data collection and analyses adhered strictly to ethics guidelines of the Association for the Study of Animal Behaviour (UK). Ethical approval for targeted leprosy camera trap surveys and faecal sample collection at CNP, Guinea-Bissau, was granted by the University of Exeter, UK. The Institute for Biodiversity and Protected Areas in Guinea-Bissau approved and collaborated directly on all aspects of this research. Ethical approval for the work by the Taï Chimpanzee Project at TNP was given by the Ethics Commission of the Max Planck Society. The Centre Suisse de Recherches Scientifiques en Côte d'Ivoire collaborates on the research at TNP.

For human participants, this study was carried out under the ethical consent of the World Health Organization Global Leprosy Program surveillance network. All human participants gave written informed consent in accordance with the Declaration of Helsinki.

## Reporting summary

Further information on research design is available in the Nature Research Reporting Summary linked to this paper.

## Data availability

Sequence data are available from the National Center for Biotechnology Information Sequence Read Archive, BioProject (PRJNA664360) and BioSample (16207289–16207321). BioSample codes for all samples used in this study are given in the Supplementary Data. Other relevant data are available in the Article and its Supplementary Information.

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

**Acknowledgements** The work at CNP was supported by the Darwin Initiative (grant number, 26-018) and the Halpin Trust (UK) to K.J.H. and C.B.; the Fundação para a Ciência e a Tecnologia (FCT), Portugal (FCT EXPL/IVC-ANT/0997/2013; IF/01128/2014) to K.J.H.; and a GCCA+ grant (Áreas Protegidas e Resiliência às Mudanças Climáticas) financed by the European Union to A.R. and A.R.S. E.B. was partly supported by an Oxford Brookes University studentship and grants from Mohamed bin Zayed Species Conservation Fund, Primate Society of Great Britain, International Primatological Society Conservation, Conservation International/Global Wildlife Conservation Primate Action Fund (Ms Constance Roosevelt) and Primate Conservation, Inc. J.B. was supported by a doctoral grant from FCT (SFRH/BD/108185/2015) and the Boise Trust Fund (University of Oxford). M.R. was supported by a studentship funded by the NERC GW4+ Doctoral Training Partnership and the College of Life and Environmental Sciences of the University of Exeter. The work at TNP was supported by the Max Planck Society, which has provided core funding for the Taï Chimpanzee Project since 1997; the work at TNP and all analyses performed on nonhuman primate samples were supported by the German Research Council projects LE1813/10-2, WI2637/4-2 and WI 2637/3-1 within the research group FOR2136 (Sociality and Health in Primates) and LE1813/14-1 (Great Ape Health in Tropical Africa); the ARCUS Foundation grant G-PGM-1807-2491; and the Robert Koch Institute. Work was partly carried out under the Global Health Protection Programme supported by the Federal Ministry of Health on the basis of a decision by the German Bundestag. A part of this Article represents a chapter in the PhD thesis of B.M. who was supported through the Robert Koch Institute's PhD programme. The work on human specimens was supported by the Fondation Raoul Follereau (S.T.C. and C.R.J.), the Heiser Program of the New York Community Trust for Research in Leprosy (J.S.S. and C.A., grant numbers P18-000250) and the Association de Chimiothérapie Anti-Infectieuse of the Société Française de Microbiologie (C.A.). C.A. was also supported by a non-stipendiary European Molecular Biology Organization long-term fellowship (ALTF 1086-2018) and the European Union's Horizon 2020 research and innovation programme under the Marie Skłodowska-Curie grant no. 845479. We thank staff at the Instituto da Biodiversidade e das Areas Protegidas for their permission to conduct research in Guinea-Bissau and for logistical support; Q. Quecuta, Director of CNP and research assistants and local guards, in particular M. Cassamá, I.T. Camará, D. Indjai, S. Sila, A. Camará, I. Galiza, F. Ndafa, A. Camará and B.S. Vieira for assisting with data collection and providing advice in CNP; village chiefs, Nalu leaders and Régulos for granting us permission to conduct research; staff at the Ministère

de l'Enseignement Supérieur et de la Recherche Scientifique and the Ministère des Eaux et Forêts in Côte d'Ivoire and the Office Ivoirien des Parcs et Réserves for permitting the research at TNP; staff of the Taï Chimpanzee Project and the Centre Suisse de Recherches Scientifiques for support of our work in TNP; S. Lemoine for providing shapefiles on chimpanzee home ranges at TNP; and J. McKinney, N. Dhar, S. Balharry, B. Mangeat and staff at the Gene Expression Core Facility from the Ecole Polytechnique Fédérale de Lausanne for support. The authors thank B. Godley and A. Frazão-Moreira for support in developing research ideas and logistics in Guinea-Bissau; A. Stone for constructive comments on the manuscript; and all the patients and clinical staff who participated in the study.

**Author contributions** Collection and analysis of chimpanzee camera trap data (CNP) was provided by K.J.H., E.B., J.B., M.R., A.R. and A.R.S. Collection of long-term data on chimpanzees and their health (TNP) was provided by R.M.W. and F.H.L. Collection of chimpanzee samples (CNP and TNP) was provided by K.J.H., E.B., J.B., M.R., K.P., A.D., J.E.J., S.M. and F.H.L. Logistics and fieldwork was provided by K.J.H., A.R., A.R.S., E.C.-H., R.M.W. and F.H.L. Necropsies were performed by S.M., A.D., J.E.J. and F.H.L. Histopathological analyses were performed by K.M.-R. Collection and provision of human samples was provided by M.G., R.C.J., M.K., S.O.S., S.T.C. and H.Z. DNA extraction, library preparation, enrichment and whole-genome sequencing was conducted by B.M., C.A., P.B, I.M., V.J.S and M.U. PCR and SNP data were confirmed by B.M., L.V.P. and C.A. Chimpanzee microsatellite analysis was performed by C.B. and J.L.-J.

Computational analysis was performed by C.A., S.C.-S. and B.M. Dating analysis was provided by A.B. Serological investigations were conducted by B.M. and L.V.P. J.S.S. provided material and the protocol for serological investigation. Funding was acquired by K.J.H., C.A., V.J.S., J.S.S., C.B., S.T.C., R.M.W. and F.H.L. K.J.H., C.A., C.B., S.C.S. and F.H.L. wrote the manuscript with considerable input from B.M., K.P., E.B., J.B., M.R., A.D. and L.V.P., with contributions from all authors. All authors approved the submitted manuscript.

**Funding** Open access funding provided by Robert Koch-Institut.

**Competing interests** The authors declare no competing interests.

**Additional information**
**Correspondence and requests for materials** should be addressed to Fabian H. Leendertz.

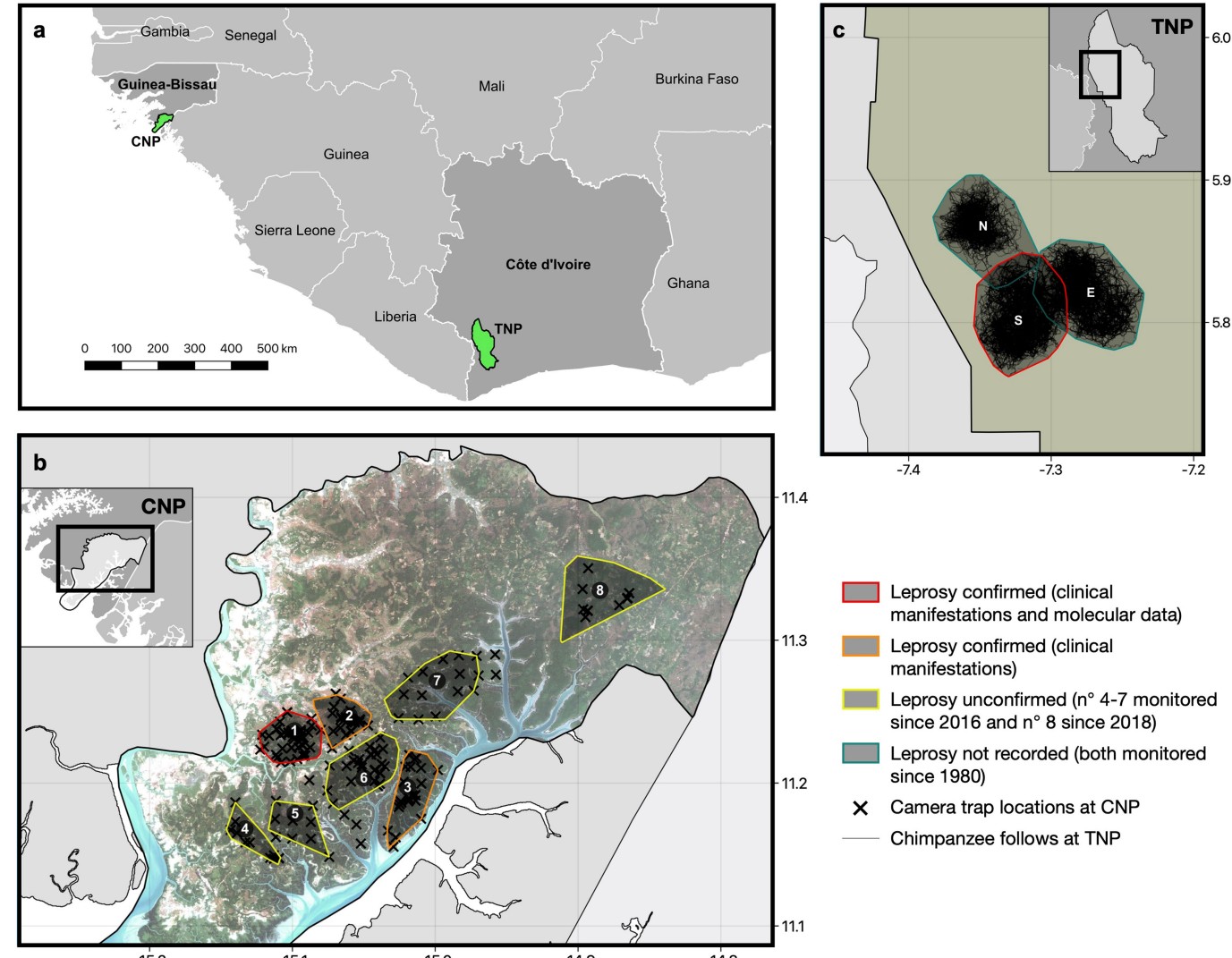

Extended Data Fig. 1 | **Maps of the chimpanzee study sites and chimpanzee communities. a**, Map of the CNP, Guinea-Bissau and the TNP, Côte d'Ivoire, West Africa. **b**, Location of the chimpanzee communities at CNP that were monitored between 2015 and 2019 (1, Caiquene-Cadique; 2, Lautchande; 3, Cambeque; 4, Cabante; 5, Canamine; 6, Madina; 7, Amindara; 8, Guiledje). Estimated home ranges of chimpanzee communities at CNP are shown by 100% minimum convex polygons of direct chimpanzee observations and indirect chimpanzee traces and nests during the study period. Red outline represents chimpanzee communities with at least one individual with clinical manifestations of leprosy, confirmed using molecular analysis; orange outline represents chimpanzee communities with at least one individual with clinical

manifestations of leprosy; yellow colour represents monitored communities where clinical manifestations of leprosy have not been observed nor confirmed through molecular analysis. **c**, Location of the three habituated chimpanzee communities monitored at TNP (N, north; S, south; E, east). Estimated home ranges of chimpanzee communities at TNP are shown by 100% minimum convex polygons of direct chimpanzee follows from December 2013 to October 2016. Red outline represents the community with individuals with clinical manifestations of leprosy, confirmed using molecular analysis and serological tests; blue colour represents communities where leprosy has not been recorded.

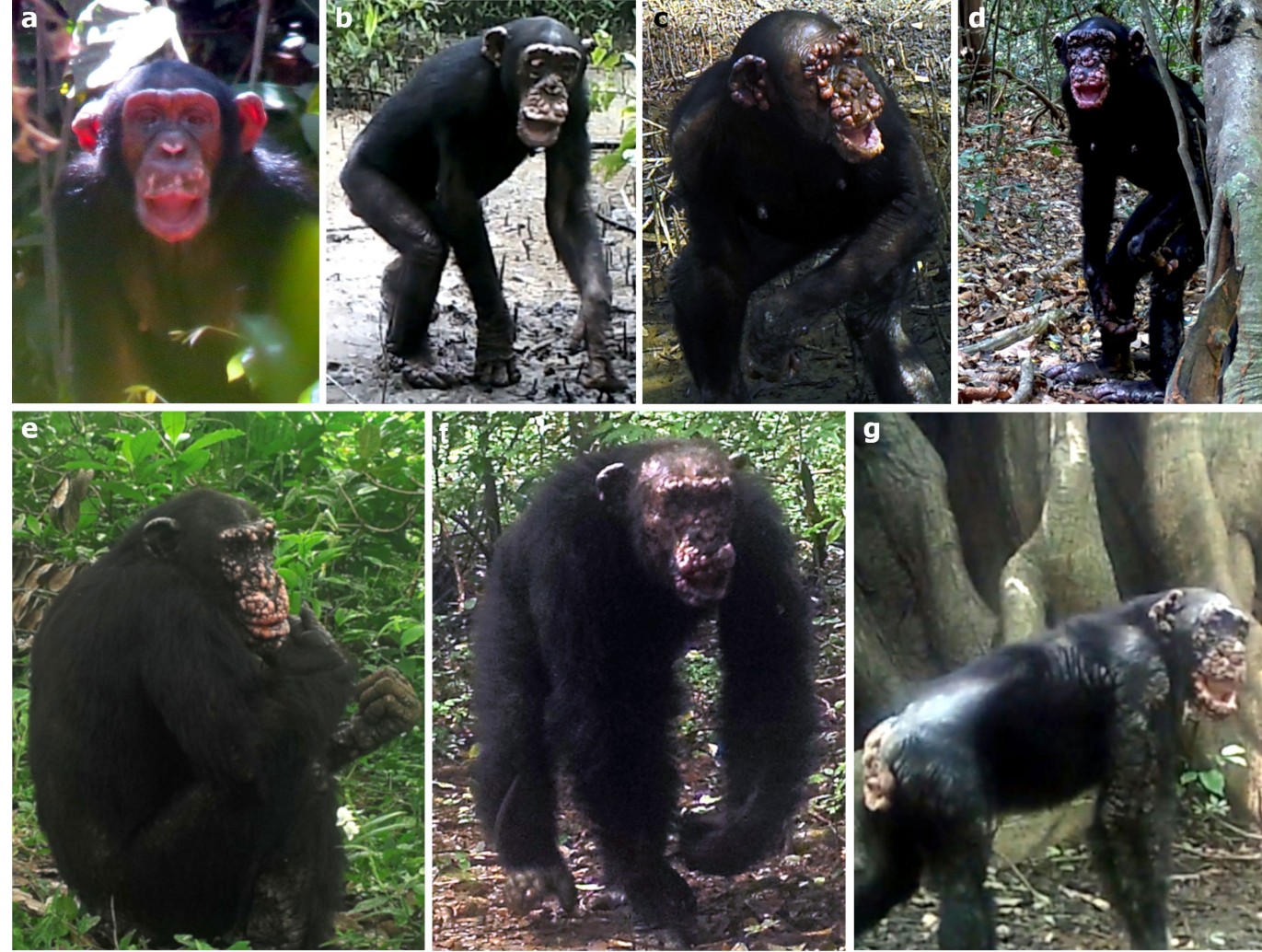

**Extended Data Fig. 2 | Disease progression of leprosy in chimpanzees at CNP.** Adult female chimpanzee Rita over the course of 5 years (**a**–**d**) and disease manifestations in three additional adult chimpanzees (**e**–**g**). **a**, 2013/05 – Hypopigmentation of skin around the mouth and nose, small nodule on the lower lip and left ear (opportunistically recorded with a video camera before the start of longitudinal health monitoring with camera traps). **b**, 2015/12 – Large nodules between the upper lip and nose, with multiple small nodules on the eyelids, cheek, ears margins, lower lip and brow ridge. Small dry patches with hair loss on the wrists, knees and elbows. **c**, 2017/12 – Nodules increase in number, with apparent swelling and reddening, facial disfigurement and claw hand. Plaques appear on the wrist, knee and elbow joints, with an increase in hair thinning. **d**, 2018/05 – Face and ears completely covered by large nodules, with facial disfigurement and generalized hair loss on limbs and lower back. Nodule formation and swelling of fingers and toes, with disfigurement of hands and feet, and more severe claw hand. Some plaques on the body are ulcerated, and the individual has clear weight loss. **e**, Jimi (Lautchande in 2018/06) – First observation of lesions in 2015. The head is completely covered with multiple nodules of reddish colour, some of which are ulcerated. Ear margins are thickened. Hands and feet present nodules and plaques, and the scrotum is affected (not visible on picture). **f**, Baaba (Cambeque in 2017/08) – First observation of lesions in 2017. Multiple hypopigmented nodules on the brow ridge, cheek and upper and lower lips. Ears have thickened margins and nodules. There is hair thinning, with multiple small plaques present on the upper and lower limbs, back, abdomen and shoulders. **g**, Brinkos (Caiquene-Cadique in 2018/10) – First observation of lesions in 2015. Facial disfigurement, with the ulceration of nodules and a hanging lower lip. Hands and feet are ulcerated, and fingers are swollen. There are nodules on the nipples, and plaques covering the lower back, shoulders and arm are ulcerated, with hair loss.

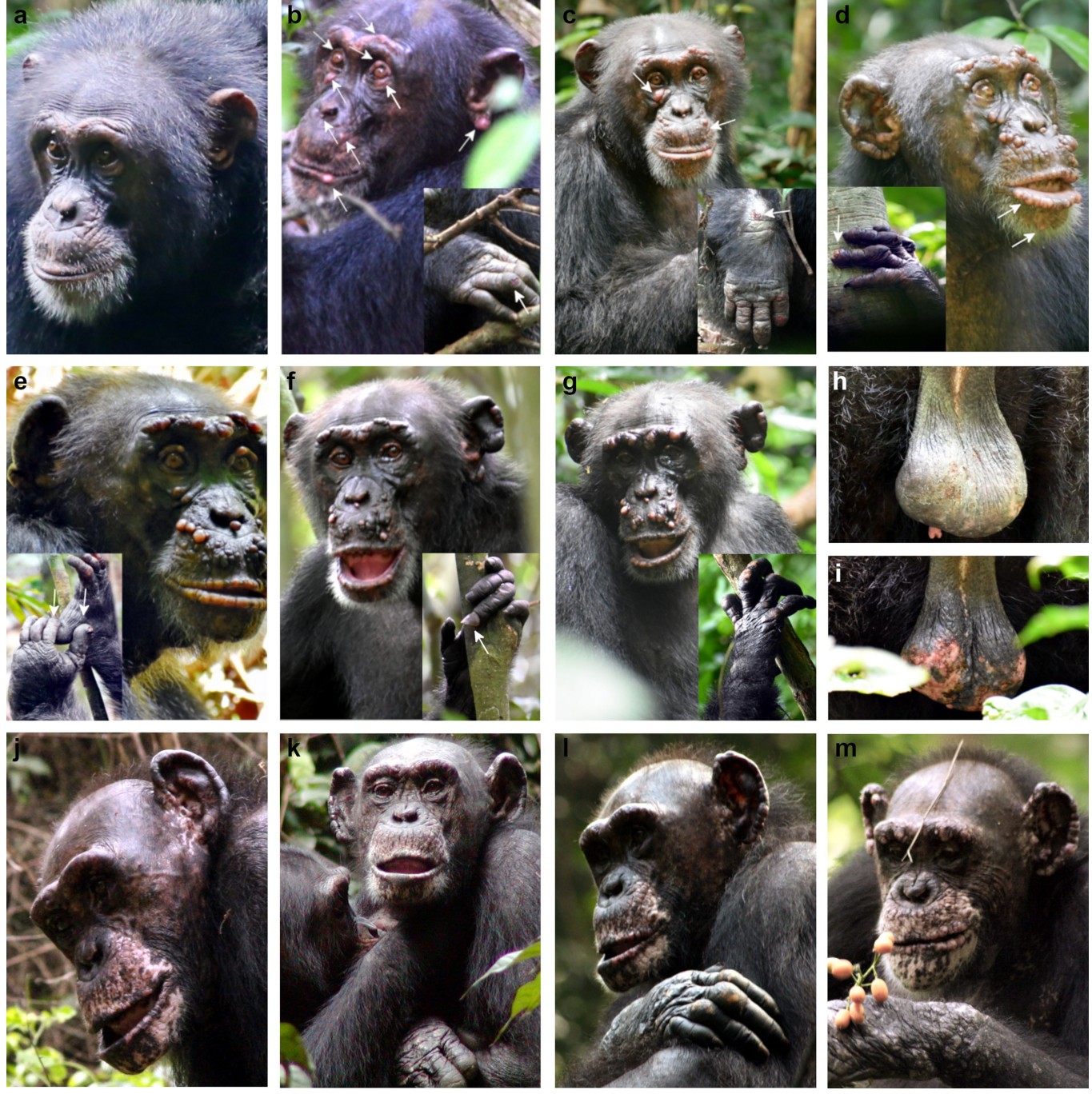

**Extended Data Fig. 3 | Disease progression of leprosy in an adult male chimpanzee at TNP (Woodstock) over the course of 2 years (2018–2020) (a–i) and an adult female chimpanzee at TNP (Zora) over the course of 2008–2009 (j–m). a**, 2017/01 – Woodstock before the appearance of clinical signs. **b**, 2018/06 – First hypopigmented nodules appear on the face (arrows), with swelling and hypopigmentation on both hands, and ulceration on the right hand. **c**, 2018/10 – Existing nodules increase in size and new smaller ones appear (arrows). Development of mucopurulent discharge from the left eye, and lower eyelid is turned outward. Hair loss and ulceration on dorsal part of right wrist and hand. **d**, 2019/04 – Most existing nodules increase in size and become pedunculated, and the nodule under the eye shrinks, and several new nodules appear (see arrows). Suspected start of nasal involvement, and right ear starts to become disfigured. Both hands are slightly swollen and hypopigmented, with the loss of nail plate on the fourth finger of the left hand, and the third and fifth fingers show early stage of abnormal nail overgrowth. **e**, 2019/10 – Facial lesions increase in size, and some become darkly pigmented. New lesions appear on the brow ridge, with nodules above the lips and between

the lips, and the nose becomes pedunculated. The loss of nail plate, and nail bed becomes exposed on the first and second fingers of the left hand. **f**, 2020/04 – In general, facial nodules seem smaller than before, and the nodule under the left eye disappears. On the left hand, the nail of the fourth finger shows an advanced stage of abnormal nail overgrowth, and the third and fifth fingernails show early stage of abnormal nail overgrowth. **g**, 2020/07 – Facial nodules seem larger with many hypopigmented, and both ears are swollen and disfigured. Nasal involvement becomes apparent. Both hands are swollen and hypopigmented. Skin ulcerations present on the right hand, with possible claw hand on the left hand. **h**, 2019/04 – Slight hypopigmentation of scrotum. **i**, 2020/07 – Reddening and ulceration of scrotum; fresh blood observed. **j**, 2007/12 – Zora before the appearance of clinical signs of leprosy. **k**, 2008/01 – Appearance of nodules on the right ear and both eyebrow ridges. **l**, 2008/12 – Appearance of nodules on the left ear, and ulceration of the skin at the second, third and fourth proximal interphalangeal joint level of the right hand. **m**, 2009/04 – Nodular lesions on both ears and brow ridge seem aggravated, with nodular lesions on the lips, and above the mouth (four months before death).

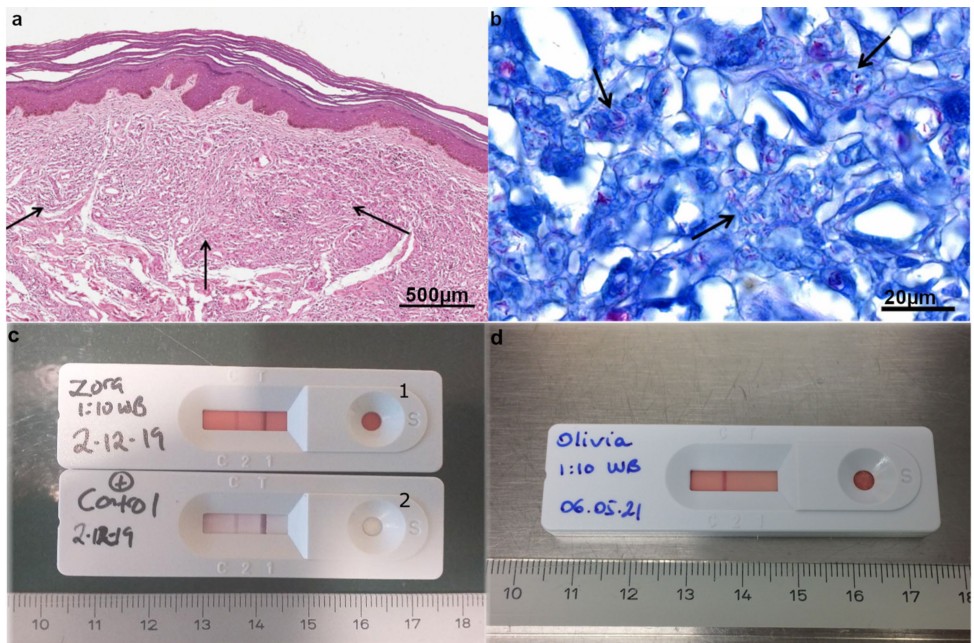

**Extended Data Fig. 4 | Confirmation of leprosy infection in Zora through histopathology of skin sample and lateral flow test. a**, Lepromatous leprosy, skin with diffuse histiocytic infiltrate in the dermis. The haematoxylin and eosin stain was conducted once; scale bar, 500 μm. **b**, Lepromatous leprosy, skin, acid-fast bacilli in histiocytes. The inflammatory infiltrate consists predominantly of histiocytes admixed with fewer lymphocytes. Histiocytes show foamy or vacuolated cytoplasm and containing bacteria surrounded by a clear zone. Fite-Faraco stain; scale bar, 20 μm. Fite-Faraco stain was conducted once and was controlled by a positive control slide containing mycobacteria. **c**, whole blood from Zora (1) and the positive control (2). **d**, whole blood from a chimpanzee at TNP (Olivia) not infected with *M. leprae*, used as negative control. C, control lane; T, test lane.

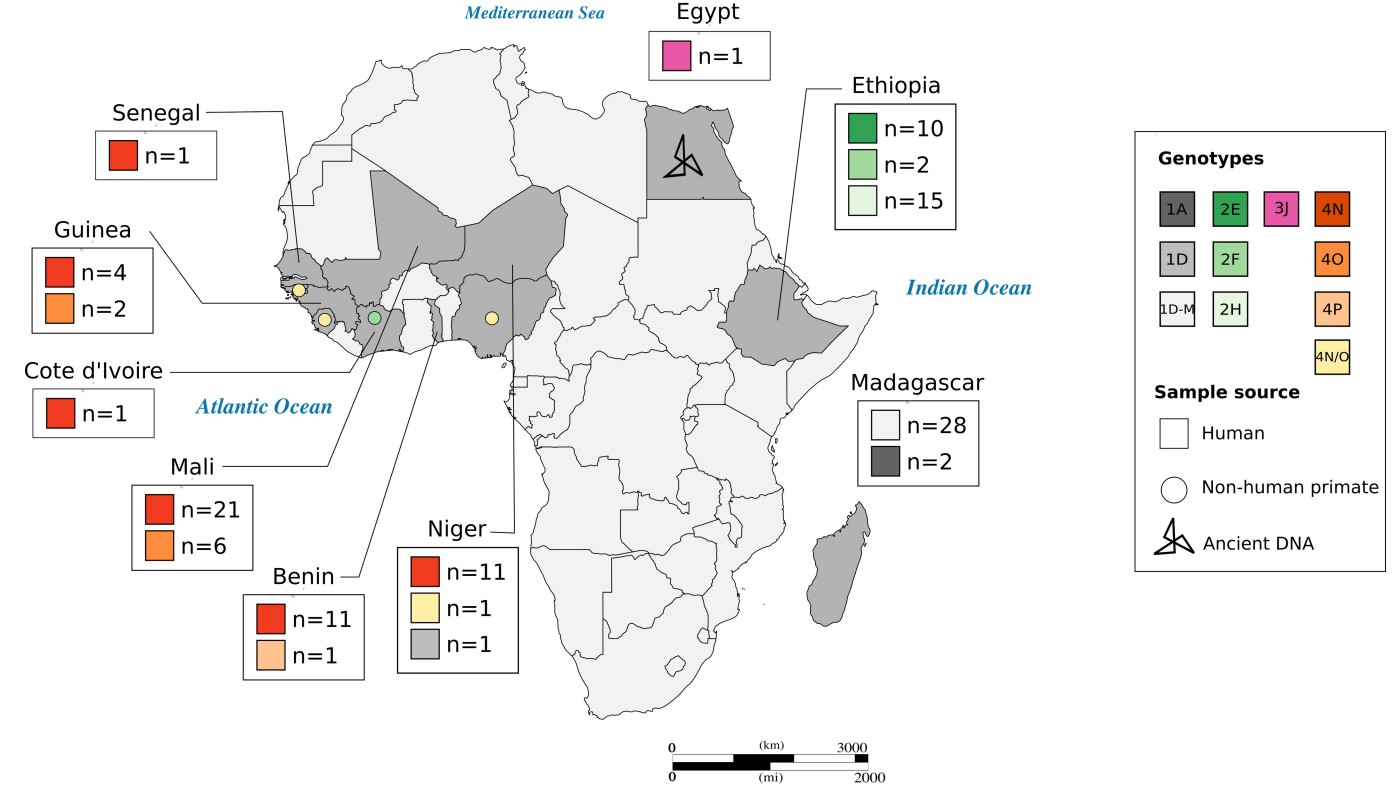

**Extended Data Fig. 5 | Geographical distribution of *M. leprae* genotypes in Africa based on genome data.** The genotype 2F has never been reported in West Africa and is the least identified in Ethiopia. The genotype 4N/O was only reported in one human sample from West Africa. Data included only *M. leprae* genomes (Supplementary Note 5 and Supplementary Table 7). The map was downloaded from https://www.amcharts.com/svg-maps/ under a free licence and modified for the current figure in Inkscape, an open source digital illustration software package (https://inkscape.org).

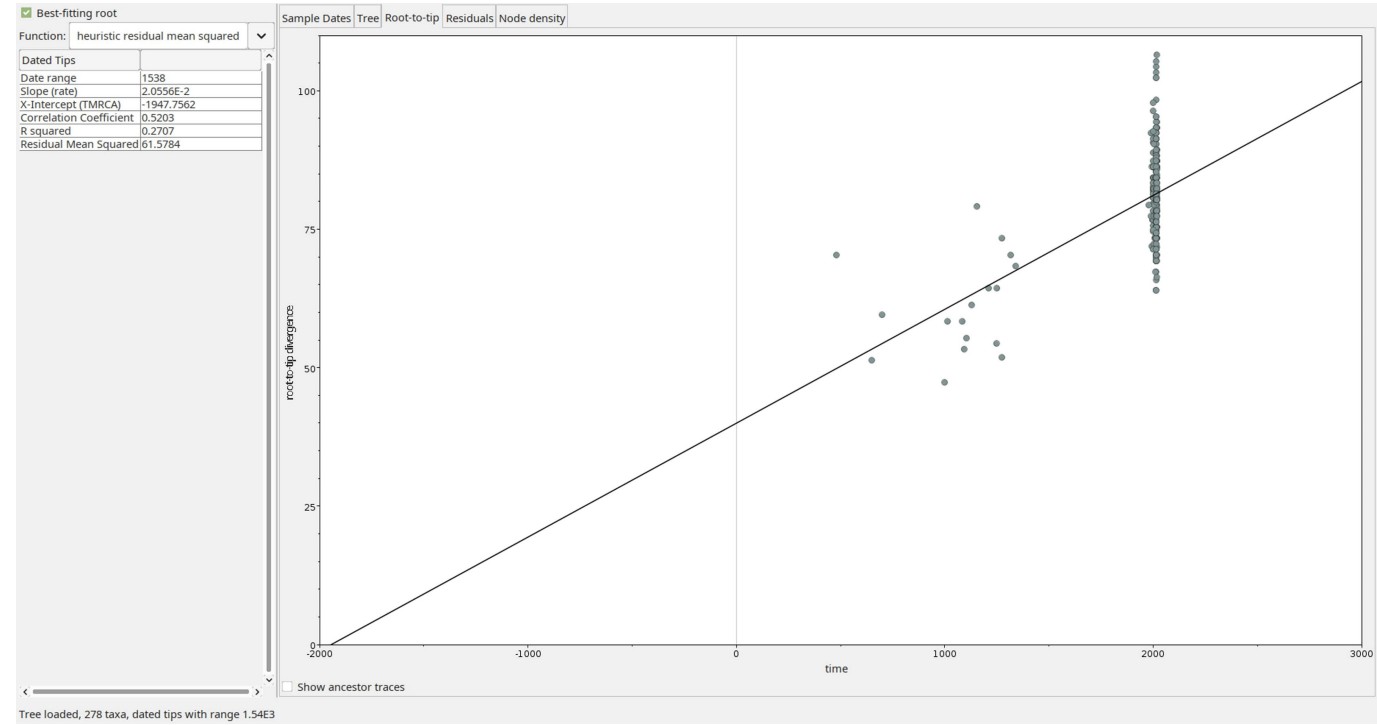

**Extended Data Fig. 6 | Best-fitting root analysis using TempEst.**

**a**

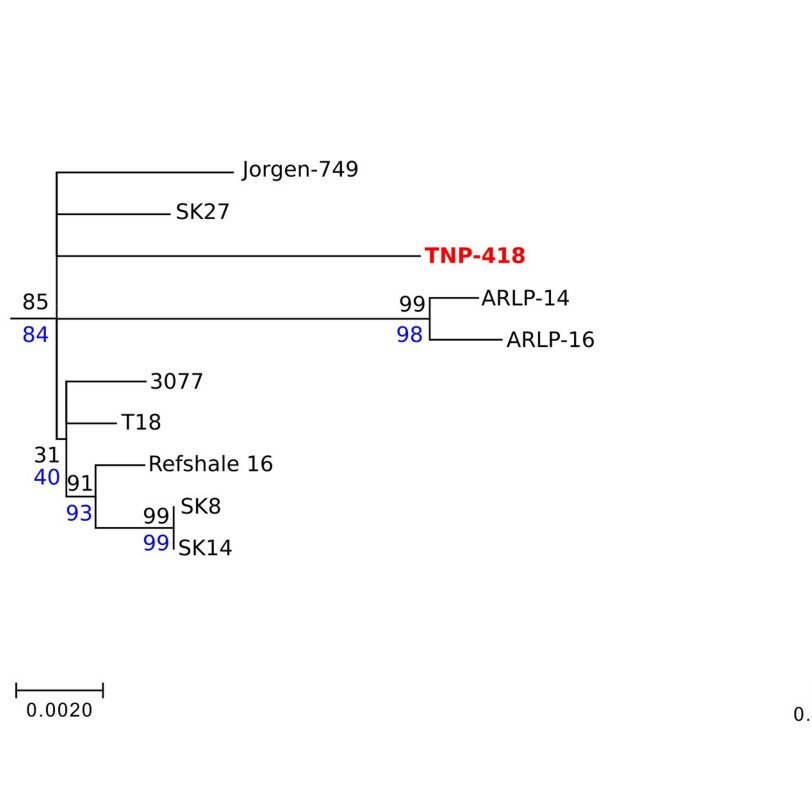

**b**

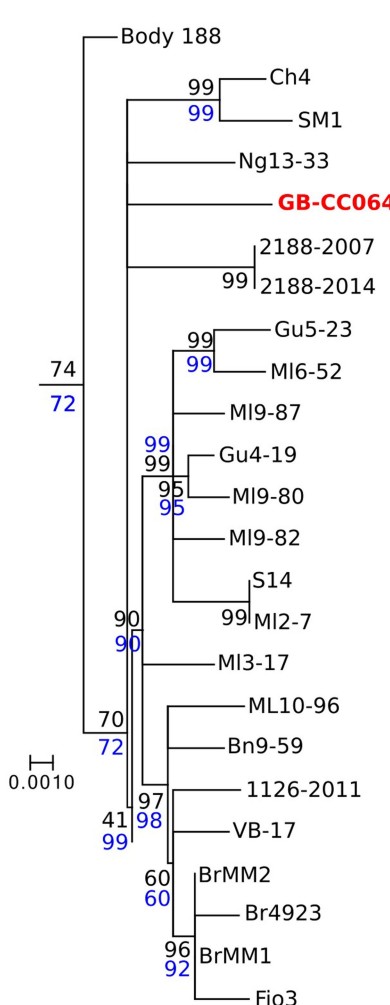

**Extended Data Fig. 7 | Maximum Likelihood tree to confirm the topological placement of GB-CC064 and TNP-418.** 286 genomes (Supplementary Table 6) were used, including the two new chimpanzee strains (in bold red), 500 bootstrap replicates (value in black with the Tamura Nei model and in blue for the general time model) and *M. lepromatosis* as outgroup. Sites with missing data were partially deleted (80% genome coverage cutoff). **a**, Maximum Likelihood tree of the branch 2F. **b**, Maximum Likelihood tree of the branch 4.

**Extended Data Table 1 | The camera trap (CT) study periods with the focal chimpanzee community within Cantanhez National Park**

| Study period | Chimpanzee community | No. CT locations | CT placement | Mode | Start date | End date | CT days | Researcher |
|---|---|---|---|---|---|---|---|---|
| 1 | Caiquene-Cadique Lautchande (2) | 21 | Targeted | Hybrid | 13.09.15 | 16.12.15 | 984 | JB EB |
| 2 | Caiquene-Cadique Lautchande Cambeque Madina Cabante Canamine-Cafache Amindara (7) | 63 | Systematic | Photo | 17.10.16 | 05.03.17 | 3237 | EB |
| 3 | Caiquene-Cadique Lautchande Cambeque Madina Cabante Canamine-Cafache (6) | 50 | Systematic | Photo | 03.06.17 | 15.11.17 | 4435 | EB |
| 4 | Caiquene-Cadique (1) | 21 | Systematic | Photo | 09.07.17 | 05.07.18 | 6838 | EB |
| 5 | Caiquene-Cadique Lautchande Cambeque Madina (4) | 52 | Targeted | Video | 20.02.17 | 08.07.18 | 8023 | JB |
| 6 | Caiquene-Cadique Lautchande Cambeque Madina Cabante Guiledje (6) | 86 | Targeted & Systematic | Video | 03.07.18 | 14.04.19 | 5476 | MR JB |

The number of distinct CT locations for that study period is included (total number of CT locations = 211). Certain CT locations were used in more than one study period (Supplementary Table 1). For targeted CT placement if no chimpanzees were filmed for a certain period CTs were repositioned; hence not all cameras were working at the same time. The placement design of CTs was targeted or systematic. Targeted CTs were deployed to maximize detection of chimpanzees (such as chimpanzee drumming sites, fruiting trees and trails). Systematic CTs were placed following a survey design maximizing independence between CT sites and chimpanzee detection. The CT mode was either set to photograph or video or both (hybrid) and CTs were active for 24 h per day. The start and end dates of each study period are included as well as the number of CT days. CT days are the sum of number of days for each active CT after removing days when cameras were inactive due to malfunctioning, batteries running out, trees falling in front of the CT or theft (total CT days = 28,993). The researcher initials are included (J.B., Joana Bessa; E.B., Elena Bersacola; M.R., Marina Ramon).

**Extended Data Table 2 | Samples (CNP) and animals (TNP) that tested positive for *M. leprae* DNA**

| Sample or animal ID | Sampling date | Sample type (*n* samples) | *M. leprae* PCR results (*n* RLEP, *n* 18kDa) | Mean depth of coverage of *M. leprae* genomes | *M. leprae* genotype |
|---|---|---|---|---|---|
| GB-CC064 | 10 - 11/2017 | Faeces (n=1) | Positive (1, 1) | 39.3X | 4N/O |
| GB-CC068 | 10 - 11/2017 | Faeces (n=1) | Positive (1, 0) | Not tested | Not tested |
| Woodstock | 01/2009 | Faeces (n=1) | Positive (1, 1) | 1.1X | 2F |
|  | 02/2009-03/2017 | Faeces (n=13) | Negative |  |  |
|  | 06/2018-01/2019 | Faeces (n=12) | Positive (12, 11) |  |  |
| Zora | 03/2001-03/2002 | Faeces (n=7) | Negative | 25.8X | 2F |
|  | 06/2002-07/2003 | Faeces (n=3) | Positive (3, 0) |  |  |
|  | 12/2003.01/2004 | Faeces (n=2) | Negative |  |  |
|  | 03/2004-05/2004 | Faeces (n=3) | Positive (3, 0) |  |  |
|  | 06/2004-08/2004 | Faeces (n=2) | Negative |  |  |
|  | 10/2004-09/2005 | Faeces (n=4) | Positive (4, 2) |  |  |
|  | 12/2005 | Faeces (n=1) | Negative |  |  |
|  | 02/2006-06/2009 | Faeces (n=16) | Positive (16, 12) |  |  |
|  | 08/2009 | Tissue (n=1) | Negative |  |  |
|  |  | Tissue (n=8) | Positive (8, 5) |  |  |
| D'Artagnan | 04/2009 | Tissue (n=10) | Negative | Not tested | Not tested |
|  |  | Tissue (n=6) | Positive (6, 0) |  |  |
| Utan | 01/2009 | Faeces (n=1) | Positive (1, 1) | Not tested | Not tested |
|  | 03/2009-07/2009 | Faeces (n=3) | Negative |  |  |
| Sagu | 01/2009-05/2009 | Faeces (n=19) | Negative (n=19) | Not tested | Not tested |

Samples are considered as PCR-positive if they were positive either in the RLEP or in the RLEP and the 18-kDa assay (see Supplementary Table 2 for additional information).

**Extended Data Table 3 | Primers used for the identification of *M. leprae* in chimpanzee tissues and faeces, diet analysis, the genotyping of *M. leprae* strains and confirmation of chimpanzee origin of the samples**

| PCR system and target | Primer pair 5´-3´ | Product size (bp) | Annealing temperature (°C) |
|---|---|---|---|
| RLEP_Primary PCR | Fwd: TGCATGTCATGGCCTTGAGG<br><br>Rev: CACCGATACCAGCGGCAGAA | 129 | 58 |
| RLEP_Nested PCR | Fwd:TGAGGTGTCGGCGTGGTC<br><br>Rev: CAGAAATGGTGCAAGGGA | 99 | 58 |
| Fusion_M13_RLEP PCR | Fwd: GTAAAACGACGGCCAGTGAGGTGTCGGCGTGGTC<br><br>Rev: CAGGAAACAGCTATGACCAGAAATGGTGCAAGGGA | 139 | 58 |
| 18kDA _Primary PCR | Fwd: TCATAGATGCCTAATCGACTG<br><br>Rev: GGCACATCTGCGGCCAGCA | 136 | 55 |
| 18kDA _Nested PCR | Fwd: ATCGACTGTTGTTTGCGCAAC<br><br>Rev: CCAGCAACCGAAATGTTCGGA | 110 | 55 |
| Fusion_M13_18kDA PCR | Fwd:GTAAAACGACGGCCAGATCGACTGTTGTTTGCGCAAC<br><br>Rev: CAGGAAACAGCTATGACCCAGCAACCGAAATGTTCGGA | 150 | 55 |
| 16S Mammal identification PCR (diet analysis) | 16Smam1: CGGTTGGGGTGACCTCGGA<br><br>16Smam2: GCTGTTATCCCTAGGGTAACT<br><br>16Smam_Human_blocker: CGGTTGGGGCGACCTCGGAGCAGAACCC<br><br>16Smam_Pig_blocker: CGGTTGGGGTGACCTCGGAGTACAAAAAAC | 130 | 64 |
| Mammal identification fusion PCR (diet analysis) | 16Smam1_Illumina_adapter:<br>TCGTCGGCAGCGTCAGATGTGTATAAGAGACAGCGGTTGGGGTGACCTCGG<br><br>16Smam2_Illumina_adapter:<br>GTCTCGTGGGCTCGGAGATGTGTATAAGAGACAGGCTGTTATCCCTAGGGTAACT | 160 | 64 |
| 16S Species confirmation PCR | 16Smam1: CGGTTGGGGTGACCTCGGA<br><br>16Smam4: AGATAGAAACCGACCTGGAT | 300 | 64 |
| Genotyping of leprosy strain infecting Woodstock | ml0048-Fwd*: ATACCGTGACGCGGATAAAC<br><br>ml0048-Rev*: GTAGCCAGTCCAAGGCAATC | 576 | 55 |
| | ml0565-Fwd**: AGCTGAGGTTGACCTGGAA<br><br>ml0565-Rev**: GTAGATTGGCGTCGTCAAAA | 561 | 57 |

Fwd, forward; Rev, reverse. *Mutation C1193T in *ml0048* (genome position 60123); a T is found in TNP-418 and TNP-566. **Mutation C319T in *ml0565* (genome position 683097); a T is found in TNP-418 and TNP-566.

# nature research

# Reporting Summary

Nature Research wishes to improve the reproducibility of the work that we publish. This form provides structure for consistency and transparency in reporting. For further information on Nature Research policies, see Authors & Referees and the Editorial Policy Checklist.

## Statistics

For all statistical analyses, confirm that the following items are present in the figure legend, table legend, main text, or Methods section.

| n/a | Confirmed | |
|---|---|---|
| ☐ | ☒ | The exact sample size (*n*) for each experimental group/condition, given as a discrete number and unit of measurement |
| ☐ | ☒ | A statement on whether measurements were taken from distinct samples or whether the same sample was measured repeatedly |
| ☒ | ☐ | The statistical test(s) used AND whether they are one- or two-sided<br>*Only common tests should be described solely by name; describe more complex techniques in the Methods section.* |
| ☐ | ☒ | A description of all covariates tested |
| ☐ | ☒ | A description of any assumptions or corrections, such as tests of normality and adjustment for multiple comparisons |
| ☐ | ☒ | A full description of the statistical parameters including central tendency (e.g. means) or other basic estimates (e.g. regression coefficient) AND variation (e.g. standard deviation) or associated estimates of uncertainty (e.g. confidence intervals) |
| ☒ | ☐ | For null hypothesis testing, the test statistic (e.g. *F*, *t*, *r*) with confidence intervals, effect sizes, degrees of freedom and *P* value noted<br>*Give P values as exact values whenever suitable.* |
| ☐ | ☒ | For Bayesian analysis, information on the choice of priors and Markov chain Monte Carlo settings |
| ☒ | ☐ | For hierarchical and complex designs, identification of the appropriate level for tests and full reporting of outcomes |
| ☒ | ☐ | Estimates of effect sizes (e.g. Cohen's *d*, Pearson's *r*), indicating how they were calculated |

*Our web collection on statistics for biologists contains articles on many of the points above.*

## Software and code

Policy information about availability of computer code

| Data collection | No software was used |
|---|---|
| Data analysis | All raw reads were adapter- and quality-trimmed with Trimmomatic v0.35. The quality settings were "SLIDINGWINDOW:5:15 MINLEN:40". Paired-end (PE) data were additionally processed with SeqPrep (https://github.com/jstjohn/SeqPrep) to merge overlapping pairs. Preprocessed reads were mapped onto the M. leprae TN reference genome (GenBank AL450380.1) with Bowtie2 v2.2.5. SNP calling was done using VarScan v2.3.9. To avoid false-positive SNP calls the following cutoffs were applied: minimum overall coverage of five non-duplicated reads, minimum of three non-duplicated reads supporting the SNP, mapping quality score >8, base quality score >15, and a SNP frequency above 80%. InDel calling was done using Platypus v0.8.1 followed by manual curation. We used the Integrative Genomics Viewer v 2.8.13 and Basic Local Alignment Search Tool (BLAST) v 2.11.0+. Dating analyses were done using BEAST2 v2.5.2. |

For manuscripts utilizing custom algorithms or software that are central to the research but not yet described in published literature, software must be made available to editors/reviewers. We strongly encourage code deposition in a community repository (e.g. GitHub). See the Nature Research guidelines for submitting code & software for further information.

## Data

Policy information about availability of data

All manuscripts must include a data availability statement. This statement should provide the following information, where applicable:

- Accession codes, unique identifiers, or web links for publicly available datasets
- A list of figures that have associated raw data
- A description of any restrictions on data availability

Sequence data are available from the NCBI Sequence Read Archive (SRA) Bioproject PRJNA664360 Biosamples SAM16207289-16207321. Biosample codes for all

# Field-specific reporting

Please select the one below that is the best fit for your research. If you are not sure, read the appropriate sections before making your selection.

☐ Life sciences ☐ Behavioural & social sciences ☒ Ecological, evolutionary & environmental sciences

For a reference copy of the document with all sections, see nature.com/documents/nr-reporting-summary-flat.pdf

# Ecological, evolutionary & environmental sciences study design

All studies must disclose on these points even when the disclosure is negative.

| | |
|---|---|
| Study description | Study description – We report on leprosy-like lesions in two wild populations of western chimpanzees in the Cantanhez National Park (CNP), Guinea-Bissau, and the Taï National Park (TNP), Côte d'Ivoire, West Africa. We screen chimpanzee faecal and necropsy samples for the presence of M. leprae and conduct phylogenomic comparisons with other strains from humans and other animals. |
| Research sample | The research sample is represented by two populations of wild chimpanzees (Pan troglodytes verus) in CNP and TNP. We conducted this study in these two populations in response to leprosy-like lesions observed during behavioural monitoring. We did not discriminate between age and sex classes, instead we collected data on as many individuals as possible for analysis of leprosy symptoms. Analyses in this paper focus on symptomatic individuals. These two chimpanzee populations include male and female individuals and age estimates range from newborn to adult (~40 years of age). There are a minimum of 12 chimpanzee communities at CNP, all unhabituated to researchers, with approximately 35-60 individuals per community (age and sex composition of all communities unknown). At one community (Caiquene-Cadique), we estimate at least 48 individuals, including 16 adult females, 13 adult males, 3 subadults and 16 immatures (juveniles and infants). At TNP, the three human-habituated chimpanzee communities include a total of 91 individually recognised chimpanzees. |
| Sampling strategy | We performed non-invasive sampling through the collection of faeces from symptomatic and asymptomatic chimpanzees at CNP and TNP. In CNP, where chimpanzees are not habituated to human observers, this is performed by collecting faecal material found under chimpanzee nests or in proximity to chimpanzee signs (e.g. food remains or knuckle prints). At the time of faecal collection, the identity of the chimpanzee was not known. At CNP, camera traps were deployed at 211 locations including across different habitat types within the home range of eight of the 12 putative chimpanzee communities. Targeted camera traps were deployed to record and monitor chimpanzee behaviour and disease occurrence. Systematic camera traps were deployed across central CNP at a minimum distance of 1km between sampling points. At TNP chimpanzees are followed by researchers on a daily basis and faeces are collected right after observing defecation. In both cases, faeces are collected with the aid of a plastic or wooden spatula and placed in 2ml or 15ml tubes dry or with RNAlater. For this study we analysed all available faecal samples from individuals which displayed clinical signs of leprosy and optimal sample sizes could not be determined beforehand. For TNP we included only samples from the South community since leprosy was observed only in members of this chimpanzee community. Necropsies on dead chimpanzees were performed by trained veterinarians at TNP as part of the health monitoring program. For this study, we tested all available chimpanzee necropsy samples in our collection. |
| Data collection | Data collection was performed by local field assistants, researchers and veterinarians working at CNP and TNP. At CNP, clinical data on unhabituated chimpanzees were collected using camera traps and faecal samples were collected with the aid of a wooden spatula and stored at ambient temperature in 15ml tubes containing NAP buffer. At TNP, data were collected by research assistants both on paper sheets and using the Cybertracker app, and by veterinarians who documented via pictures and videos. At TNP, the long-term health monitoring program includes continuous collection of faecal and urine samples from known adult chimpanzees. Faeces are transferred in 2ml cryotubes with the aid of a plastic spatula and frozen in liquid nitrogen. A full necropsy is systematically performed on chimpanzees found dead by the on-site veterinarian. Tissue samples of several internal organs are taken if the state of carcass decomposition allows. |
| Timing and spatial scale | Camera traps were set up over six data collection periods ranging from 2015 to 2019 across CNP (1067 km2). There were six study periods in total: (1) 13.09.15-16.12.15 (984 camera trap (CT) days, targeted CT placement, 2 communities); (2) 17.10.16-05.03.17 (3237 CT days, systematic, 7 communities); (3) 03.06.17-15.11.17 (4435 CT days, systematic, 6 communities); (4) 09.07.17-05.07.18 (6838 CT days, systematic, 1 community); (5) 20.02.17-08.07.18 (8023 CT days, targeted, 4 communities); (6) 03.07.18-14.04.19 (5476 CT days, targeted and systematic, 6 communities). Data collection was stopped once we had obtained sufficient camera trap footage to determine leprosy presence across chimpanzee communities. Since 2020, the Cantanhez Chimpanzee Project has continued monitoring the health of this population. At TNP sample collection for the project started in 1994 and has been routinely carried out ever since. Over 25 years we have accumulated a collection of chimpanzee faecal and urine samples and necropsy samples from all wildlife found dead in the area. For this study, we tested samples collected between 1998 and 2019. |
| Data exclusions | No specifica data were excluded from the study. |
| Reproducibility | To confirm our results of leprosy infection we used two PCR systems in parallel and tested several samples for each individual/community. Positives were then further confirmed via next generation sequencing. For this purpose several individual libraries were generated to confirm M. leprae DNA presence in the samples. |
| Randomization | Randomization is not relevant for this type of study, which is based on investigating infectious causes of illness in wildlife. To maximize our chances of pathogen detection we sampled all individuals, whenever possible. |

| | Blinding | Not applicable to this study since this is a study on a naturally occurring disease in wild animals. |
|---|---|---|

Did the study involve field work?  ☒ Yes  ☐ No

## Field work, collection and transport

| Field conditions | Guinea-Bissau (36,125 km2), West Africa, lies within the Guinean forest-savannah mosaics, a biodiverse ecoregion buffering the Guinean moist forests in the south and the West Sudanian savannah in the north. The climate in Guinea-Bissau is characterized by a rainy season from mid-May to the end of October and a long dry season from November to mid-May. Cantanhez NP (N11° 14.287' W15° 02.281') comprises the Cubucaré peninsula in the Tombali Region bordering Guinea-Conakry. The landscape in Cantanhez NP consists of a mosaic of coastal sub-humid forest patches, mangroves, savannah grassland, woodland and agriculture including mostly cashew orchards, shifting cultivation fields and mangrove swamp rice fields. Approximately 24,000 people across 200 villages and settlements are present inside the park. The TNP (5,082 km 2), located in the south-west of Ivory Coast bordering Liberia (N5° 38 56 W7° 05 43), consists of an evergreen lowland rainforest and is the largest remaining primary forest fragment in West Africa. It is home to a wide range of mammals that include 11 different nonhuman primate species. There are no settlements or agricultural areas inside the National Park. The climate in TNP is characterized by a rainy season from March/April to the end of October and a dry season from November to February/March. |
|---|---|
| Location | Tai National Park, Ivory Coast and Cantanhez National Park, Guinea Bissau |
| Access and import/export | Research conducted at CNP is authorised by the Institute for Biodiversty and Protected Areas (IBAP) in Guinea-Bissau, who are partners and co-authors on this research. All research at TNP is conducted under the umbrella of a collaboration with Ivorian partners and health authorities. Samples are routinely exported to Germany for diagnostic purposes following international guidelines and prior official authorization through CITES permits, where necessary. CITES permits for importing necropsy samples from Ivory Coast are regulary issued to the RKI. The most recent ones were issued on March 30th 2021 under the number DE-E-05895/20 and DE-E-05896/20. |
| Disturbance | All activities conducted for this study were carried out as part of the Cantanhez Chimpanzee Project and the Tai Chimpanzee Project. All samples and observations collected are done with the minimum disturbance to wildlife and the environment. At CNP, camera traps are used to collect data and cause minimum disturbance to chimpanzees. Faecal samples are collected when animals are no longer at the site. At TNP, a minimum distance of 7 meters is maintained from chimpanzees and samples are collected after the animals have moved away. Only non-invasive samples such as faeces and urine are collected. |

# Reporting for specific materials, systems and methods

We require information from authors about some types of materials, experimental systems and methods used in many studies. Here, indicate whether each material, system or method listed is relevant to your study. If you are not sure if a list item applies to your research, read the appropriate section before selecting a response.

## Materials & experimental systems

| n/a | Involved in the study |
|---|---|
| ☒ | Antibodies |
| ☒ | Eukaryotic cell lines |
| ☒ | Palaeontology |
| ☐ | ☒ Animals and other organisms |
| ☐ | ☒ Human research participants |
| ☒ | Clinical data |

## Methods

| n/a | Involved in the study |
|---|---|
| ☒ | ChIP-seq |
| ☒ | Flow cytometry |
| ☒ | MRI-based neuroimaging |

## Animals and other organisms

Policy information about studies involving animals; ARRIVE guidelines recommended for reporting animal research

| Laboratory animals | This study did not involve laboratory animals. |
|---|---|
| Wild animals | At CNP, chimpanzees are not habituated to human observers and all data are collected remotely using camera traps. The age and sex distribution of chimpanzees within this population have not be calculated (as this requires accurately identifying all individuals). At one community (Caiquene-Cadique), we estimate at least 48 individuals, including 16 adult females, 13 adult males, 3 subadults and 16 immatures (juveniles and infants). At TNP, wild chimpanzee communities have been habituated by researchers since 1979. A team of field assistants and researchers follow the animals on a daily basis from a 7-meter distance, recording behavioural data and collecting faeces and urine samples whenever possible. In normal situations, each assistant or researcher has one focal individual per day to collect data and samples from. In disease outbreak situations, monitoring efforts are reinforced and sampling is attempted from all symptomatic and asymptomatic individuals. These populations include male and female individuals and estimation of age range is from newborn to adult (~40 years of age). As of March 2021, there are 91 individuals (40 males and 51 females), including 43 adults (14 males and 29 females), 5 adolescents (4 males, and 1 female), 19 juveniles (6 males and 13 females), and 24 infants (16 males and 8 females). |
| Field-collected samples | At CNP, chimpanzee faecal samples are collected by visiting chimpanzee nesting and feeding sites. Faecal samples were stored at room temperature in 15ml tubes containing NAP buffer, and shipped to Robert Koch Institute in Germany. At TNP, samples are |

collected upon defecation or urination of the chimpanzees and stored in 2ml cryotubes. The research camps of the Tai Chimpanzee Project are equipped with liquid nitrogen tanks for storage of samples. Samples are then transported to Abidjan for temporary storage at the Centre Suisse de Recherches Scientifiques and subsequently shipped to RKI on dry ice whenever someone is traveling. Since these samples were collected from wild living animals, no other parameter needs to be specified (e.g. housing or photoperiod).

| | |
|---|---|
| Ethics oversight | All data were collected in accordance with Best Practise Disease and Monitoring Guidelines of the Great Ape Section of IUCN Primate Specialist Group. The collection of samples was strictly non-invasive. All proposed data collection and analyses adhered strictly to ethics guidelines of the Association for the Study of Animal Behaviour (UK). Ethical approval for targeted leprosy camera trap surveys and faecal sample collection at CNP, Guinea-Bissau, was granted by the University of Exeter, UK. The Institute for Biodiversity and Protected Areas (IBAP) in Guinea-Bissau approved and collaborated directly on all aspects of this research. Ethical approval for the work at Tai Chimpanzee Project was given by the Ethics Commission of the Max Planck Society. The Centre Suisse de Recherches Scientifiques en Côte d'Ivoire collaborates on the research at TNP. |

Note that full information on the approval of the study protocol must also be provided in the manuscript.

# Human research participants

Policy information about studies involving human research participants

| | |
|---|---|
| Population characteristics | M. leprae strains were collected from skin samples of newly diagnosed patients with positive bacillary index. These were obtained from the respective National Leprosy Control Programs in the framework of the leprosy drug resistance surveillance programs. Among the 21 patients included retrospectively in this study, seven were female and 13 were male (one unknown), ranging from 18 to 80 years in age. They originated from Mali (n=8), Benin (n=6), Niger (n=5), Côte d'Ivoire (n=1) and Senegal (n=1). |
| Recruitment | Patients were not recruited for this study. Inform consent were collected by the respective National Leprosy Control Programs during diagnosis to allow the use of the M. leprae strain genetic informations. |
| Ethics oversight | This study was carried out under the ethical consent of the WHO Global Leprosy Program surveillance network. All subjects gave written informed consent in accordance with the Declaration of Helsinki. |

Note that full information on the approval of the study protocol must also be provided in the manuscript.

