## [Peer Review File · Nature]

Manuscript Title: Leprosy in wild chimpanzees

Editorial Notes:

Reviewer Comments & Author Rebuttals

Reviewer Reports on the Initial Version:

Referee #1 (Remarks to the Author):

I have concentrated my review on the phylogenetic analysis, as that is where my expertise is. The analysis performed is robust and based on sound and up-to-date techniques. I have no concerns about this.

However, I do take issue with the final conclusion of “multiple introductions of *M. leprae* from an unknown animal or environmental source” in the abstract and “sporadic emergence from environmental sources” at the end of the discussion.

This seems very unlikely, given the phylogeny. Although the chimpanzee isolates are basal to two separate modern human groups, all of them, including the chimpanzee isolates, are well within the diversity of modern human *M. leprae*. As this has an ancestor only 3,000 years ago, the strongest hypothesis has to be that this represents a single ancient introduction into the human population, as indicated by previous literature on *M. leprae* phylogenomics. In addition, both chimpanzee isolates branch among ancient human samples. Therefore, whatever the source for the introductions of these strains into chimpanzees, they must have previously come from humans at some point.

“Sporadic emergence” from the environment implies to me that there is an environmental reservoir, unrelated to human infection, from which these chimpanzee strains have come, and the phylogeny provides no evidence for this at all. It provides even less evidence that there is another “main reservoir” of *M. leprae* outside of humans. Support for both of these statements would require much stronger evidence, including genome sequences for non-human or environmental isolates outside of or basal to the current tree, which are not provided. These conclusions therefore should be removed or substantially toned down.

Referee #2 (Remarks to the Author):

This manuscript describes the identification of *Mycobacterium leprosy* in wild populations of chimpanzees in Cantanhez National Park (CNP), Guinea-Bissau and Tai National Park (TNP), Côte d'Ivoire. While *M. leprae* has been seen as typically human-restricted, long-standing evidence has shown that it can infect and sustain in armadillo populations. Red squirrels in Britain and Ireland have also more recently been revealed as maintenance hosts of both *M. leprae* and the related *M. lepromatosis*. In this work the authors provide further evidence for the surprisingly wide host range of leprosy bacilli. With a multidisciplinary approach that includes observational studies, pathological analyses, and molecular and genomic characterisation, they reveal *M. leprae* can sustain in populations of chimpanzees. Furthermore, their work suggests that other animal hosts or environmental reservoirs of *M. leprae* may exist.

The difficulty of studying such wild populations is considerable. The current COVID-19 pandemic has reinforced the importance of understanding the diversity of infectious agents that may exist in animal populations. Towards this, the work reported in this manuscript shows how critically important it is to study wildlife populations for infections in their natural habitat using minimally

invasive approaches.

In general, I found the manuscript to be clearly written and easy to follow. I do however have some comments I would like the authors to address.

Comments

1. In terms of the evidence presented, the data from CNP is that 2 samples gave PCR positivity for *M. leprae*, one of which (GB-CC064) gave sufficient DNA to allow for genome sequencing, revealing the rare 4 N/O genotype. On this latter point, it is not clear to me whether the evidence to rule out recent human transmission is that strong. While I understand that the 4 N/O genotype is rare in human samples that have been sequenced to date, perhaps this rarity is more a function of not having the sample depth of human samples from Guinea Bissau? Indeed, as the authors state, "Although chimpanzees at CNP inhabit an agroforest landscape and share access to natural and cultivated resources with humans, present-day human-chimpanzee direct contact is uncommon. The exact nature of historic human-chimpanzee interactions at CNP remains, however, unknown." While I of course understand that missing data often bedevils such studies, I would welcome further clarification on this point from the authors.

The evidence from TNP in terms of long-term maintenance of *M. leprae* infection is stronger. The relative isolation of the TNP animals from humans appears more absolute than for the CNP animals, and would suggest that this is not 'spill-over' of recent human infection. Furthermore, the depth of samples available in biobanks allowed a detailed retrospective analysis for evidence of infection going back to 2009. The PCR results over time from the same animals is a major bonus. The fact that sufficient DNA could be recovered from Zora for WGS provides further confirmatory data and allows the *M. leprae* isolates to be placed in the context of the wider global phylogeny and temporal evolution of *M. leprae*.

2. In the abstract (lines 59 to 60) it states "...but spill-over to other mammals such as nine banded armadillos and red squirrels occurs". I would rephrase this sentence as it suggests that spill-over of infection is happening currently into nine banded armadillos and red squirrels. However, it would be more correct to view armadillos and red squirrels as maintenance hosts that have been infected for decades/centuries. Hence, a different phrasing could be "...but spill-over has occurred to other mammals that are now maintenance hosts, such as nine banded armadillos and red squirrels."

3. Line 92-94: "In captivity, nonhuman primates, such as chimpanzees (), sooty mangabeys () and cynomolgus macaques (), developed leprosy spontaneously (i.e. not through laboratory experiments)." I would rephrase this as it is somewhat confusing. Perhaps a better phrasing would be "In captivity, nonhuman primates (...) have been known to develop leprosy without any obvious infectious source. Due to their captive status..."

4. Table 1: As well as just showing the over PCR positive/negative results, it would have been good to see a breakdown of those samples that were either positive for RELEP or for both RELEP and the 18 kDa gene (as supp data).

5. Line 240: "...and beyond within the group comprising all genotype 4". I'm unclear what "beyond within the group" means here.

6. Line 313 "also show that *M. leprae* multiplies in amoebae, arthropods, and ticks,". Could the authors provide a reference for the replication of *M. leprae* in ticks? Is it Ferreira et al, 2018 PLOS NTD?

7. Line 315 "...our findings challenge the long-held assumption that humans are the main reservoir of *M. leprae* and suggest that this pathogen may sporadically emerge from environmental

sources." I don't believe that the authors have challenged that humans are the main reservoir of *M. leprae*. Indeed, as the authors state themselves, at least for TNP "it seems leprosy is a rare disease with low transmission levels in these chimpanzee communities". Perhaps a more qualified conclusion would be that their "findings reveal new maintenance hosts of *M. leprae* and suggest that this pathogen may sporadically emerge from as yet unidentified environmental sources".

8. Extended data Figure 6. The figure shows two lateral flow results, one with blood from Zora the other with a positive control. However, I would have thought that a necessary negative control to show would have been a LFA with blood from an animal not infected with *M. leprae*. Could the authors add this?

Referee #3 (Remarks to the Author):

This is a review for "Leprosy in wild chimpanzees". The authors identify leprosy in wild chimpanzees from two different locations. They are able to recover genetic material for some of them and to whole or partial genome sequence the leprosy pathogen. Overall the study raises the major intriguing question about the real host range of leprosy, potential reservoirs and how leprosy has evolved in wild fauna with little contact with humans. The manuscript is clearly written, the conclusions are robust and hypothesis well-funded and possible alternatives to explain the observation are formulated in the discussion. There are some questions I would like the authors to answer/address although most of them are minor and in the direction to reinforce the message:

1. While I understand that you use specific bait enrichment, how likely is you are sequencing another closely related mycobacteria and only keeping the information common to that mycobacteria and the *M. leprae*? Have you analyzed the content of non-mapped read to see if there are reads indicative of other Mycobacteria?. It is not very likely but again, because of the special characteristics of these populations it maybe happen that another mycobacteria (known or unknown) is causing similar lesions? For example I assume that *M. lepromatosis*, while rare, will have a very closely genome that can be (occasionally) amplified by your approach. Looking at the results I think it is a remote possibility but running a taxon composition analysis will reinforce the point that there is no other mycobacteria around and you are in a unique possibility to address this. This should be accompanied by the % of reads assigned to *Mleprae* to have an idea on how complex in terms of composition was the initial sample.
2. In general there is some confusion because of the inconsistent use of the term coverage. Looks like to me that sometimes the authors refer to genome coverage and others to depth of coverage (or fold coverage). Can the authors look at this and correct if needed so is more clear for the reader?
3. The hypothesis that the human-chimp transmission is unlikely is very plausible and perfectly explained by the data and the authors. In that regard and additional point you can make is that there is no marker of drug resistance detected in those genomes. First, I would like to know if this is true (I presume yes) and second you maybe interested to add this point in the discussion.
4. L295 - you mention that the human population density 2000 ya makes unlikely any human-> animal transmission. However the habitat of those chimpanzees may have changed over time and they may have arrived to their current location later. Can you elaborate on ancestral chimpanzees habitats and/or potential historical movements? Any chance this may reconnect the strains types observed with those closely matching in the phylogeny?
5. Fig 3. I assume the bayesian phylogeny presented is that obtained from BEAST? For clarity I would add to the captio: Bayesian dated phylogenetic tree. The BEAST bayesian phylogenies are somewhat special as they incorporate time as well as other parameters not explicitly modeled in more common bayesian phylogenetic analysis that does not incorporate dating. So I think is better to mention that is coming from a dating analysis.
6. In fact there are some topological differences with the parsimony phylogeny (Fig3b,c). Related to this and given 1) The low diversity in the alignment (percentage of invariants very high) and

that 2) parsimony does not deal well (or incorporate explicitly) missing data I would like the authors to check that the relationships holds true using a ML framework. I recognize that is very unlikely a change in the scenario but given that the manuscript is presenting few samples, recovered with baits, I think it is worth introducing a method that can deal better with missing data.

7. I presume there is very little clock signal for dating in *M. leprae* but I have not seen a correlation between time and genetic distance. Can the authors elaborate on this? I am aware we are talking more of a general issue (and some of the authors have already published other dating works) than for your manuscript but It will be nice to add some discussion points about this in the main text or supplementary, as a limitation or not

Author Rebuttals to Initial Comments:

Referee #1

- I have concentrated my review on the phylogenetic analysis, as that is where my expertise is. The analysis performed is robust and based on sound and up-to-date techniques. I have no concerns about this.

We are glad that the reviewer supports the methods and analyses employed.

- However, I do take issue with the final conclusion of “multiple introductions of *M. leprae* from an unknown animal or environmental source” in the abstract and “sporadic emergence from environmental sources” at the end of the discussion. This seems very unlikely, given the phylogeny. Although the chimpanzee isolates are basal to two separate modern human groups, all of them, including the chimpanzee isolates, are well within the diversity of modern human *M. leprae*. As this has an ancestor only 3,000 years ago, the strongest hypothesis has to be that this represents a single ancient introduction into the human population, as indicated by previous literature on *M. leprae* phylogenomics. In addition, both chimpanzee isolates branch among ancient human samples. Therefore, whatever the source for the introductions of these strains into chimpanzees, they must have previously come from humans at some point. “Sporadic emergence” from the environment implies to me that there is an environmental reservoir, unrelated to human infection, from which these chimpanzee stains have come, and the phylogeny provides no evidence for this at all. It provides even less evidence that there is another “main reservoir” of *M. leprae* outside of humans. Support for both of these statements would require much stronger evidence, including genome sequences for non-human or environmental isolates outside of or basal to the current tree, which are not provided. These conclusions therefore should be removed or substantially toned down.

*We have reworked the abstract and discussion and have substantially toned down our conclusions. Until we have additional data on *M. leprae* in other nonhuman animal and environmental sources it is hasty to state the most likely explanation for our data is an environmental source, unrelated to humans. We have now clarified that a human source at both sites is impossible to rule out. We have revised the discussion to make it clear that humans are the most likely source of leprosy in chimpanzees at CNP due to human-chimpanzee coexistence across an agro-forest matrix. We have extended our discussion to better describe the situation at TNP where chimpanzees are more distant from local human communities. Please see below for responses to Reviewers 2 and 3 for revisions made to similar comments.*

Referee #2

- This manuscript describes the identification of *Mycobacterium leprosy* in wild populations of chimpanzees in Cantanhez National Park (CNP), Guinea-Bissau and Taï National Park (TNP), Côte d'Ivoire. While *M. leprae* has been seen as typically human-restricted, long-standing evidence has shown that it can infect and sustain in armadillo populations. Red squirrels in Britain and Ireland have also more recently been revealed as maintenance hosts of both *M. leprae* and the related *M. lepromatosis*. In this work the authors provide further evidence for the surprisingly wide host range of leprosy bacilli. With a multidisciplinary approach that includes observational studies, pathological analyses, and molecular and genomic characterisation, they reveal *M. leprae* can sustain in populations of chimpanzees. Furthermore, their work suggests that other animal hosts or environmental reservoirs of *M. leprae* may exist. The difficulty of studying such wild populations is considerable. The current COVID-19 pandemic has reinforced the importance of understanding the diversity of infectious agents that may exist in animal populations. Towards this, the work reported in this manuscript shows how critically important it is to study wildlife populations for infections in their natural habitat using minimally invasive approaches. In general, I found the manuscript to be clearly written and easy to follow.

We are grateful that the reviewer appreciates the critical importance of examining animal health in the wild using non-invasive methods, and finds the manuscript clearly written.

- In terms of the evidence presented, the data from CNP is that 2 samples gave PCR positivity for *M. leprae*, one of which (GB-CC064) gave sufficient DNA to allow for genome sequencing, revealing the rare 4 N/O genotype. On this latter point, it is not clear to me whether the evidence to rule out recent human transmission is that strong. While I understand that the 4 N/O genotype is rare in human samples that have been sequenced to date, perhaps this rarity is more a function of not having the sample depth of human samples from Guinea Bissau? Indeed, as the authors state, "Although chimpanzees at CNP inhabit an agroforest landscape and share access to natural and cultivated resources with humans, present-day human-chimpanzee direct contact is uncommon. The exact nature of historic human-chimpanzee interactions at CNP remains, however, unknown." While I of course understand that missing data often bedevils such studies, I would welcome further clarification on this point from the authors. The evidence from TNP in terms of long-term maintenance of *M. leprae* infection is stronger. The relative isolation of the TNP animals from humans appears more absolute than for the CNP animals, and would suggest that this is not 'spill-over' of recent human infection. Furthermore, the depth of samples available in biobanks allowed a detailed retrospective analysis for evidence of infection going back to 2009. The PCR results over time from the same animals is a major bonus. The fact that sufficient DNA could be recovered from Zora for WGS provides further confirmatory data and allows the *M. leprae* isolates to be placed in the context of the wider global phylogeny and temporal evolution of *M. leprae*.

*We have reworked the discussion and now state that “The exact nature of historic human-chimpanzee interactions at CNP remains, however, unknown. For example, data on whether chimpanzees used to be kept as pets or were hunted for meat are currently lacking. The long-term coexistence between humans and chimpanzees in this shared landscape makes humans the most probable source of chimpanzee infection. However, multiple individuals from several chimpanzee communities across Cantanhez NP show symptomatic leprosy demonstrating that *M. leprae* is now likely transmitted between individuals within this population.” (lines 243-252 in unmarked version). We agree that more concrete social and historical data on historic human-chimpanzee interactions will enable us to determine the human and/or chimpanzee behaviours that might have resulted in *M. leprae* transmission from humans to chimpanzees.*

- In the abstract (lines 59 to 60) it states “...but spill-over to other mammals such as nine banded armadillos and red squirrels occurs”. I would rephrase this sentence as it suggests that spill-over of infection is happening currently into nine banded armadillos and red squirrels. However, it would be more correct to view armadillos and red squirrels as maintenance hosts that have been infected for decades/centuries. Hence, a different phrasing could be “...but spill-over has occurred to other mammals that are now maintenance hosts, such as nine banded armadillos and red squirrels.”

Thank you for the suggestion, we have made the requested change to the abstract.

- Line 92-94: “In captivity, nonhuman primates, such as chimpanzees (), sooty mangabeys () and cynomolgus macaques (), developed leprosy spontaneously (i.e. not through laboratory experiments).” I would rephrase this as it is somewhat confusing. Perhaps a better phrasing would be “In captivity, nonhuman primates (...) have been known to develop leprosy without any obvious infectious source. Due to their captive status...”

Thank you for the suggestion, we have made the requested change.

- Table 1: As well as just showing the over PCR positive/negative results, it would have been good to see a breakdown of those samples that were either positive for RELEP or for both RELEP and the 18 kDa gene (as supp data).

We have updated Table 1 to include the information requested (Line 146) and also state that further information can be found in Supplementary Table 1.

- Line 240: “...and beyond within the group comprising all genotype 4”. I’m unclear what “beyond within the group” means here.

We have deleted the word ‘beyond’.

- Line 313 “also show that *M. leprae* multiplies in amoebae, arthropods, and ticks,”. Could the authors provide a reference for the replication of *M. leprae* in ticks? Is it Ferreira et al, 2018 PLOS NTD?

We now cite Ferreira et al 2018 to support our statement on the replication of M. leprae in ticks. (Line 283).

- Line 315 “...our findings challenge the long-held assumption that humans are the main reservoir of *M. leprae* and suggest that this pathogen may sporadically emerge from environmental sources.” I don’t believe that the authors have challenged that humans are the main reservoir of *M. leprae*. Indeed, as the authors state themselves, at least for TNP “it seems leprosy is a rare disease with low transmission levels in these chimpanzee communities”. Perhaps a more qualified conclusion would be that their “findings reveal new maintenance hosts of *M. leprae* and suggest that this pathogen may sporadically emerge from as yet unidentified environmental sources”.

Taking account of all reviewer suggestions, we have now adapted the concluding sentence to “Testing these hypotheses will require thorough investigation into the distribution of M. leprae in wildlife and the environment. This may shed light on the relatively mysterious transmission pathways of M. leprae.” (Lines 284-286).

- Extended Data Figure 6. The figure shows two lateral flow results, one with blood from Zora the other with a positive control. However, I would have thought that a necessary negative control to show would have been a LFA with blood from an animal not infected with *M. leprae*. Could the authors add this?

In Extended Data Figure 6, we now include a negative control from a chimpanzee not infected with leprosy. We selected an individual (Olivia) from the same group as Zora and Woodstock (the two chimpanzees from TNP with leprosy) who never showed leprosy-like lesions and whose faecal samples were negative in both PCR systems used. Blood from this chimpanzee was obtained during a necropsy performed after she succumbed to a fatal human respiratory syncytial virus infection in December 2009.

Referee #3

- This is a review for “Leprosy in wild chimpanzees”. The authors identify leprosy in wild chimpanzees from two different locations. They are able to recover genetic material for some of them and to whole or partial genome sequence the leprosy pathogen. Overall the study raises the major intriguing question about the real host range of leprosy, potential reservoirs and how leprosy has evolved in wild fauna with little contact with humans. The manuscript is clearly written, the conclusions are robust and hypothesis well-funded and possible alternatives to explain the observation are formulated in the discussion. There are some questions I would like the authors to answer/address although most of them are minor and in the direction to reinforce the message.

We are pleased the reviewer finds the manuscript well-written and robust.

- While I understand that you use specific bait enrichment, how likely is you are sequencing another closely related mycobacteria and only keeping the information common to that mycobacteria and the *M. leprae*? Have you analyzed the content of non-mapped read to see if there are reads indicative of other Mycobacteria?. It is not very likely but again, because of the special characteristics of these populations it maybe happen that another mycobacteria (known or unknown) is causing similar lesions? For example I assume that *M. lepromatosis*, while rare, will have a very close genome that can be (occasionally) amplified by your approach. Looking at the results I think it is a remote possibility but running a taxon composition analysis will reinforce the point that there is no other mycobacteria around and you are in a unique possibility to address this. This should be accompanied by the % of reads assigned to *M. leprae* to have an idea on how complex in terms of composition was the initial sample.

*The bait enrichment is likely to also capture DNA fragments from other mycobacteria similarly to the array capture that our laboratory has implemented in the past (Singh et al, 2016, PNAS, <https://doi.org/10.1073/pnas.1421504112>). Nevertheless, it is unlikely that this would go unnoticed because, for each genome sequenced, the alignment is visualized on IGV with a specific focus on *rpoB* for drug resistance purpose but also because *rpoB* is highly conserved between mycobacteria and mixed situation will be observed in IGV. For example, the *rpoB* sequence of *M. leprae* shared 94% of nucleotide identity with the sequence from *M. lepromatosis*. The *rrs* region is also systematically checked on IGV. From the random blast of a few highly dissimilar fragments, it is clear that other bacteria are also found in the faeces of GB-CC064 and TNP-418. However, this region as well as other ribosomal genes and repeats are removed prior to phylogenetic analysis to avoid possible misreconstruction of the phylogeny. Currently, *M. lepromatosis* is the closest organism genetically to *M. leprae* and shares 93% in nucleotide identity with *M. leprae* which is lower compared to pathogens from the *M. tuberculosis* complex for example (~99% nucleotide identity), or between the different genotypes of *M. leprae* (~99.99%). With the current threshold used to call SNPs, and knowing that our approach would also capture *M. lepromatosis* fragments, the presence of *M. lepromatosis* would therefore be identified in the SNP table by an abnormally high number of false SNP calls, which is not the case for TNP-418 nor GB-CC064. This would be similar for any other mycobacteria. Finally, the fact that the two *M. leprae* strains fall into two different branches which are not ancestral reinforces the results that the reads aligned in our analysis belong to *M. leprae* and not to another closely related mycobacteria. The percentage of reads assigned to *M. leprae* is shown in Extended Table 3.*

- In general there is some confusion because of the inconsistent use of the term coverage. Looks like to me that sometimes the authors refer to genome coverage and others to depth of coverage (or fold coverage). Can the authors look at this and correct if needed so is more clear for the reader?

We have now clarified when we are referring to genome coverage and fold coverage in the main manuscript, Supplementary Information and Extended Data.

- The hypothesis that the human-chimp transmission is unlikely is very plausible and perfectly explained by the data and the authors. In that regard and additional point you can make is that there is no marker of drug resistance detected in those genomes. First, I would like to

know if this is true (I presume yes) and second you maybe interested to add this point in the discussion.

There are indeed no markers of resistance as exemplified by the wild-type sequence of the drug resistance determining regions in the rpoB (rifampicin), folP1 (dapsone), gyrA/gyrB (fluoroquinolones) genes (please see Supplementary Table 3). This point has now been added to the discussion (Line 262) and to the Supplementary Information Note 6.

- L295 - you mention that the human population density 2000 ya makes unlikely any human-> animal transmission. However the habitat of those chimpanzees may have changed over time and they may have arrived to their current location later. Can you elaborate on ancestral chimpanzees habitats and/or potential historical movements? Any chance this may reconnect the strains types observed with those closely matching in the phylogeny?

Barratt et al (2020) show that the areas that have remained consistently more climatically stable over the last 120,000 years closely match the forest refugia which form part of chimpanzees current distribution. Chimpanzee distribution 2000ya (based on habitat suitability) would not have extended into Ethiopia and hence does not explain the strain type observed in chimpanzees at TNP.

Barratt et al (2020 (PREPRINT) <https://www.biorxiv.org/content/10.1101/2020.05.15.066662v2>.

- Fig 3. I assume the bayesian phylogeny presented is that obtained from BEAST? For clarity I would add to the caption: Bayesian dated phylogenetic tree. The BEAST bayesian phylogenies are somewhat special as they incorporate time as well as other parameters not explicitly modeled in more common bayesian phylogenetic analysis that does not incorporate dating. So I think is better to mention that is coming from a dating analysis.

We have made the recommended change to the legend of Figure 3. (line 418).

- In fact there are some topological differences with the parsimony phylogeny (Fig3b,c). Related to this and given 1) The low diversity in the alignment (percentage of invariants very high) and that 2) parsimony does not deal well (or incorporate explicitly) missing data I would like the authors to check that the relationships holds true using a ML framework. I recognize that is very unlikely a change in the scenario but given that the manuscript is presenting few samples, recovered with baits, I think it is worth introducing a method that can deal better with missing data.

We have now performed Maximum likelihood analysis using the same alignment used for the Maximum Parsimony using the following parameter: bootstrap 500, partial deletion 80%, using the Tree Inference option Nearest-Neighbor-Interchange and the Tamura Nei model as well as general time model. The topology obtained by ML analysis is consistent with the MP analysis for both strains from chimpanzees. The methods and results of this new analysis were added as Extended Data Figure 8 and Supplementary Information Note 5.

- I presume there is very little clock signal for dating in *M. leprae* but I have not seen a correlation between time and genetic distance. Can the authors elaborate on this? I am aware we are talking more of a general issue (and some of the authors have already published other dating works) than for your manuscript but It will be nice to add some discussion points about this in the main text or supplementary, as a limitation or not

We have now measured temporal signal using two methods TempEst and BETS. Both TempEst and BETS methods showed a temporal signal in our data, which justifies performing a dating analysis. Additional information is now included in the Supplementary Information Note 5 "Temporal signal for dating analysis in M. leprae".

Yours sincerely,

Kimberley Hockings and Fabian Leendertz (on behalf of all authors)

Reviewer Reports on the First Revision:

Referee #1 (Remarks to the Author):

I am satisfied that the authors have responded to my comments.

Referee #2 (Remarks to the Author):

The authors have satisfactorily addressed all my previous comments, and with the inclusion of the other reviewers' comments I find the manuscript to be greatly improved.

One comment I do have would be that the final sentence is a little weak (i.e. concluding with "relatively mysterious" transmission pathways). Perhaps it would be better to finish with "Testing these hypotheses will require thorough investigation into the distribution of *M. leprae* in wildlife and the environment, and so shed light on the pathogen's overall transmission pathways."

Referee #3 (Remarks to the Author):

After reading the revised manuscript I think the authors have done a good job regarding reviewers concerns.

Author Rebuttals to First Revision:

.....
Referee #2 (Remarks to the Author):

The authors have satisfactorily addressed all my previous comments, and with the inclusion of the other reviewers' comments I find the manuscript to be greatly improved. One comment I do have would be that the final sentence is a little weak (i.e. concluding with "relatively mysterious" transmission pathways). Perhaps it would be better to finish with "Testing these hypotheses will

require thorough investigation into the distribution of M. leprae in wildlife and the environment, and so shed light on the pathogen's overall transmission pathways."

We have now edited the final sentence in line with the reviewer's suggestion.

.....

Yours sincerely,

Dr Kimberley Hockings and Dr Fabian Leendertz